# CD98hc is a target for brain delivery of biotherapeutics

Kylie S. Chew[1,5], Robert C. Wells[1,5], Arash Moshkforoush[1], Darren Chan[1], Kendra J. Lechtenberg[1], Hai L. Tran [1], Johann Chow [1], Do Jin Kim[1], Yaneth Robles-Colmenares[1], Devendra B. Srivastava[1], Raymond K. Tong[1], Mabel Tong[1], Kaitlin Xa[1], Alexander Yang[1], Yinhan Zhou[1], Padma Akkapeddi[1], Lakshman Annamalai[1], Kaja Bajc[2,3], Marie Blanchette[2,3], Gerald Maxwell Cherf[1], Timothy K. Earr[1], Audrey Gill[1], David Huynh[1], David Joy [1], Kristen N. Knight[1], Diana Lac[1], Amy Wing-Sze Leung[1], Katrina W. Lexa[1], Nicholas P. D. Liau[1], Isabel Becerra[1], Mario Malfavon[2,3], Joseph McInnes[1], Hoang N. Nguyen [1], Edwin I. Lozano[1], Michelle E. Pizzo [1], Elysia Roche[1], Patricia Sacayon[1], Meredith E. K. Calvert[1], Richard Daneman[2,3], Mark S. Dennis[1], Joseph Duque[1], Kapil Gadkar[1], Joseph W. Lewcock [1], Cathal S. Mahon[1], René Meisner[1], Hilda Solanoy[1], Robert G. Thorne[1,4], Ryan J. Watts[1], Y. Joy Yu Zuchero[1]✉ & Mihalis S. Kariolis [1]✉

Brain exposure of systemically administered biotherapeutics is highly restricted by the blood-brain barrier (BBB). Here, we report the engineering and characterization of a BBB transport vehicle targeting the CD98 heavy chain (CD98hc or SLC3A2) of heterodimeric amino acid transporters (TV$^{CD98hc}$). The pharmacokinetic and biodistribution properties of a CD98hc antibody transport vehicle (ATV$^{CD98hc}$) are assessed in humanized CD98hc knock-in mice and cynomolgus monkeys. Compared to most existing BBB platforms targeting the transferrin receptor, peripherally administered ATV$^{CD98hc}$ demonstrates differentiated brain delivery with markedly slower and more prolonged kinetic properties. Specific biodistribution profiles within the brain parenchyma can be modulated by introducing Fc mutations on ATV$^{CD98hc}$ that impact FcγR engagement, changing the valency of CD98hc binding, and by altering the extent of target engagement with Fabs. Our study establishes TV$^{CD98hc}$ as a modular brain delivery platform with favorable kinetic, biodistribution, and safety properties distinct from previously reported BBB platforms.

The blood-brain barrier (BBB) plays a critical role in maintaining the proper functioning of the brain by preventing the entry of most biomolecules[1–5]. However, the BBB also restricts access to large molecule therapeutics designed to treat central nervous system (CNS) disorders[6,7]. Only 0.01–0.1% of circulating antibodies cross the BBB, which despite the recent clinical success of anti-amyloid drugs[8,9], is typically insufficient to drive the target engagement required for efficacy[10–13]. Limited BBB transport of antibodies is likely due to a

[1]Denali Therapeutics, Inc., 161 Oyster Point Blvd., South San Francisco, CA 94080, USA. [2]Department of Pharmacology, University of California San Diego, 9500 Gilman Dr., La Jolla 92093 CA, USA. [3]Department of Neurosciences, University of California San Diego, 9500 Gilman Dr., La Jolla, CA, USA. [4]Department of Pharmaceutics, University of Minnesota, Minneapolis, MN, USA. [5]These authors contributed equally: Kylie S. Chew, Robert C. Wells. ✉e-mail: zuchero@dnli.com; kariolis@dnli.com

combination of low vesicular trafficking and preferential lysosomal degradation of internalized antibody in brain endothelial cells[14], suggesting that the predominant route of entry may be via the blood-CSF barrier. Subsequent brain uptake relies mainly on diffusion; therefore, distribution may often be effectively limited to brain surfaces and the perivascular spaces[15]. Increasing brain exposure while enhancing biodistribution of biotherapeutics throughout the brain parenchyma could substantially increase target engagement and lead to further improvements in efficacy. Direct injections into the CNS or disruption of the BBB have been shown to enable higher brain concentrations of drug. However, these routes of administration are highly invasive and not conducive to repeat dosing, limiting their overall effectiveness[13]. There remains a critical need for safe, effective, and non-invasive CNS delivery platforms that can deliver a variety of biotherapeutics to the brain.

One solution that has been utilized for CNS delivery of biotherapeutics is receptor-mediated transcytosis (RMT), wherein receptors highly expressed on brain endothelial cells are leveraged as entry points into the brain[16–22]. Molecules entering the CNS via BBB-expressed receptors take advantage of the extensive vascularization of the brain to deliver drugs throughout the parenchyma. For example, molecules engineered to bind transferrin receptor (TfR), the most well-studied RMT target, can result in brain concentrations 20-30 times higher than their non-RMT binding counterparts[12,23]. Platforms engineered to bind TfR have been used to deliver various large molecule therapeutics to the brain parenchyma, including enzymes, proteins, and antibodies[12,23–33]. Translation into the clinic has been promising in recent years, providing validation that this approach can be successfully utilized for CNS therapeutics[27,34–37].

Targeting other proteins highly expressed on the BBB could enable brain delivery platforms with unique kinetics, biodistribution and safety properties optimally tailored for the treatment of distinct CNS disorders. Proteomic profiling of brain endothelial cells has been a useful method to identify potentially promising BBB targets[22,38]. One such BBB transport target is the CD98 heavy chain (CD98hc, also called 4F2, *Slc3a2*)[22]. CD98hc forms covalent heterodimers[39,40] with various large amino acid transporters, including LAT1, and is essential for trafficking these transporters to the cell surface (Fig. 1a)[41]. The expression of CD98hc on brain endothelial cells is high[42] and prior work has validated it as an RMT target in both mice[22] and cynomolgus monkey[30]. However, the kinetic and biodistribution properties of biotherapeutics engineered to bind CD98hc have not been investigated.

We have previously described the transport vehicle (TV) technology comprising an Fc domain engineered to bind TfR that enables brain delivery of biotherapeutics[12]. The TV was engineered by mutating a loop region on the Fc domain of the human IgG1 (huIgG), resulting in an RMT platform that retains the native huIgG architecture while also preserving FcRn binding. This contrasts with other brain delivery platforms that require fusion to non-native linkers or to additional domains[30,33,43,44]. The TV platform is highly modular, enabling brain delivery of Fabs, enzymes, or proteins by fusion to the modified Fc[12,28,29,31]. Multiple clinical-stage programs that leverage the TV platform are currently under evaluation as potential treatments for neurodegenerative diseases[28,29,45,46].

Here we report a brain delivery platform using a CD98hc binding transport vehicle (TV^CD98hc), developed by engineering the β-sheet surface of the CH3 domain. The resulting TVs are fused to Fabs to generate antibody transport vehicles (ATV^CD98hc) (Fig. 1a). Enhanced delivery of ATVs to brain is observed in humanized CD98hc knock-in mice and cynomolgus monkey. We assess ATV^CD98hc systemic clearance, brain uptake kinetics, CNS biodistribution, and peripheral safety after chronic dosing. We also examine how CD98hc binding valency, engagement with FcγR, and Fab binding alters these properties. We demonstrate that ATV^CD98hc molecules have clearance, uptake kinetics, and biodistribution properties that are differentiated from the previously reported TfR-binding antibody transport vehicle (ATV^TfR), further expanding the options available to leverage the TV platform for CNS uptake and biodistribution of diverse large molecule therapeutics.

## Results

### Engineering CD98hc-binding TVs
We sought to explore the potential for CD98hc-mediated brain delivery of biotherapeutics by developing CD98hc-binding TVs. Initial screening for CD98hc binders using libraries engineered on Fc loop regions[12] was unsuccessful, prompting the design of additional Fc-based libraries. A structural assessment of the Fc domain (PDB ID: 4W4O) was performed to identify solvent-exposed surface patches consisting of at least 10 amino acids. Residues important for FcRn and FcγR binding were avoided, whereas regions containing β-sheet secondary structure were favored to enable interrogation of epitopes with a diversity of biochemical and structural properties relative to loop-based designs.

One library made up of 11 amino acids forms a surface on the anti-parallel β-strands in the CH3 domain (Fig. 1b). Residues on this surface were diversified using a "limited liability" codon bias approach, a strategy that reduces the number and relative positioning of amino acids unfavorable for stability, solubility, and specificity (Supplementary Fig. 1a). The resulting library was screened using phage display against human CD98hc, and five unique variants from two distinct sequence families were recovered. One clone displaying cynomolgus monkey cross-reactive binding (TV6, Supplementary Fig. 1b) was chosen for affinity maturation as this property allows for assessment of the platform in non-human primate to enable better prediction of human biodistribution and safety. A pair of affinity maturation (AM) libraries were designed where portions of the TV6 sequence were held constant with the remainder randomized (AM1 and AM2; Fig. 1b, Supplementary Fig. 2a). Screening these libraries yielded variants with improved affinity to both human and cynomolgus CD98hc (Fig. 1c, d). A second round of affinity maturation was performed using two complementary libraries, AM3 and AM4 (Fig. 1b, Supplementary Fig. 2b). In AM3, an expanded surface was explored, while in AM4, positions found to be conserved in previous clones were fixed and the remaining sequence space was varied according to limited liability diversification. These libraries produced clones with improved or similar binding to both human and cynomolgus CD98hc, with many of the strongest binders containing an expanded surface explored in AM3 (Fig. 1c, d). The strongest binding, cynomolgus cross-reactive clone, TV6.8, was chosen for a final round of affinity maturation using 10 patch libraries, each with an additional 5-6 amino acids randomized near the engineered surface (Fig. 1b, Supplementary Fig. 2c). These libraries were selected by yeast surface display to CD98hc, yielding further improvements in cross-reactive binding (Fig. 1c, d).

Given the established importance of RMT target affinity[26,32], a set of clones with a diverse range of CD98hc binding affinities was desired. Weaker clones were rationally designed by reverting 1-4 positions to previously tested amino acids, completing a set of TV^CD98hc with affinities to human and cynomolgus CD98hc ranging from 20 nM–2.1 μM and 205 nM–6 μM, respectively (Fig. 1c). Cynomolgus CD98hc binding of individual clones was 3-10-fold weaker than human CD98hc binding (Fig. 1d). Sequence analysis across rounds of maturation revealed a severely restricted functional sequence space within the TV6 family with gains in affinity largely achieved through interface expansion and optimization of several critical positions. This limited diversity in sequence space, despite significant engineering, suggests an optimal solution has been approached for the TV6 family, and that further improvements in affinity are likely challenging to achieve.

TV clones were reformatted as ATVs by adding Fabs that bind to a small molecule not present in vivo, the hapten dinitrophenol (DNP), to

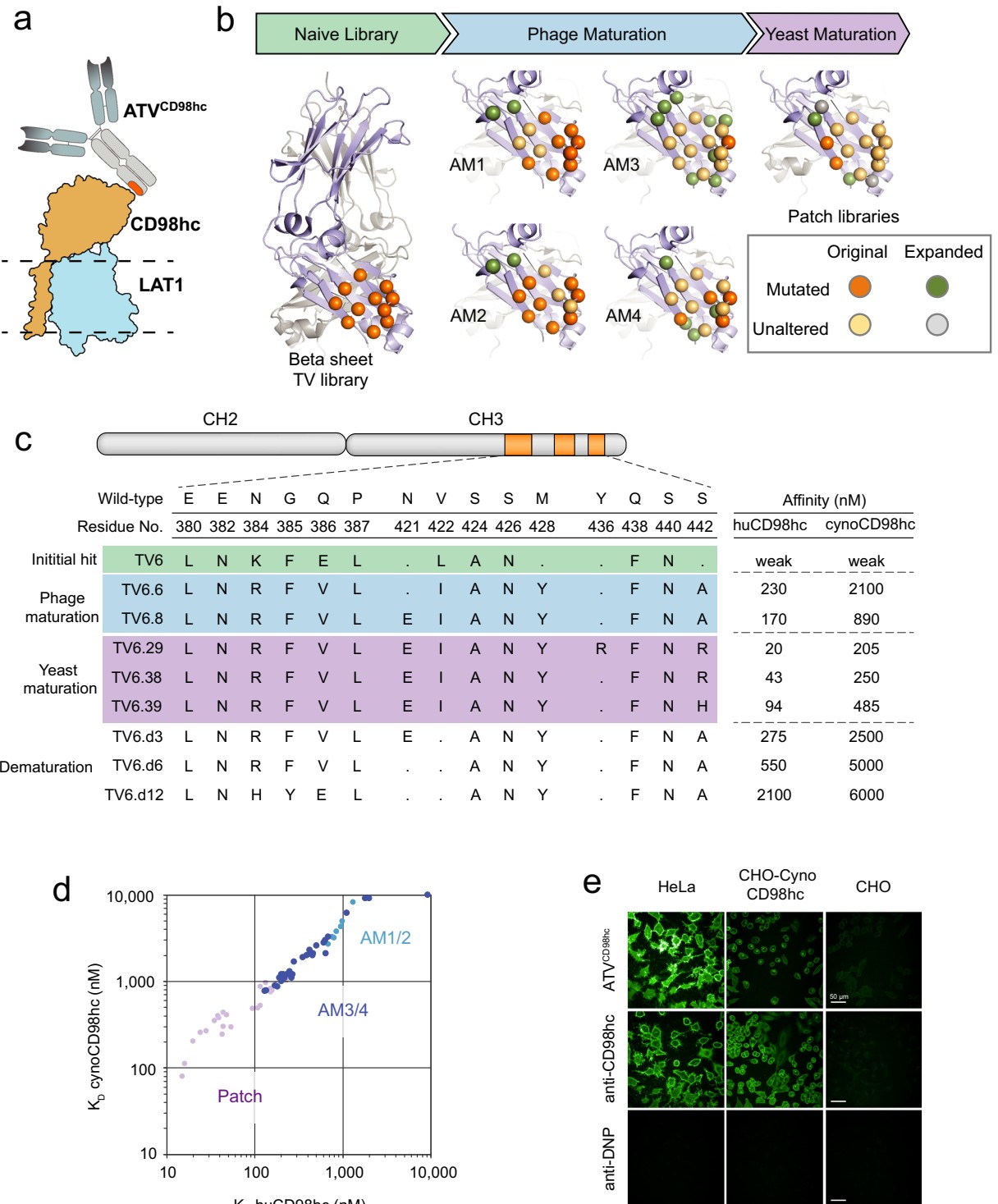

**Fig. 1 | Transport vehicles engineered at an Fc β-sheet surface bind with a range of affinities to CD98hc. a** Cartoon of the ATV$^{CD98hc}$ bound to the CD98hc-LAT1 complex with the relative position of the membrane highlighted (dotted line). **b** Outline of the steps used to identify and engineer TVs. Positions in the naïve and maturation libraries are modeled onto the human IgG1 Fc domain (PDB ID: 1HZH) with positions randomized in the original library (orange), randomized expanded positions (green), unaltered original positions (tan), and unaltered peripheral positions (gray) highlighted. An example of one of ten patch libraries is provided. **c** Representative TV sequences from each stage are shown with corresponding

affinities to human and cynomolgus CD98hc of the ATV$^{CD98hc}$:DNP clone, as measured by surface-plasmon resonance. **d** A compilation of affinities of ATV$^{CD98hc}$:DNP variants to human and cynomolgus CD98hc across rounds of maturation. **e** Immunocytochemistry detection of huIgG 1 h after incubation of ATV$^{CD98hc}$:DNP (62.5 nM), anti-CD98hc (62.5 nM) and anti-DNP (500 nM) to human (HeLa), cynomolgus CD98hc-expressing cells (CHO: cynomolgus CD98hc) and parental cells (CHO). Representative images are shown from at least 15 images per well, $n = 2$ independent experiments. Scale bars = 50 μm. Source data are provided as a Source Data file.

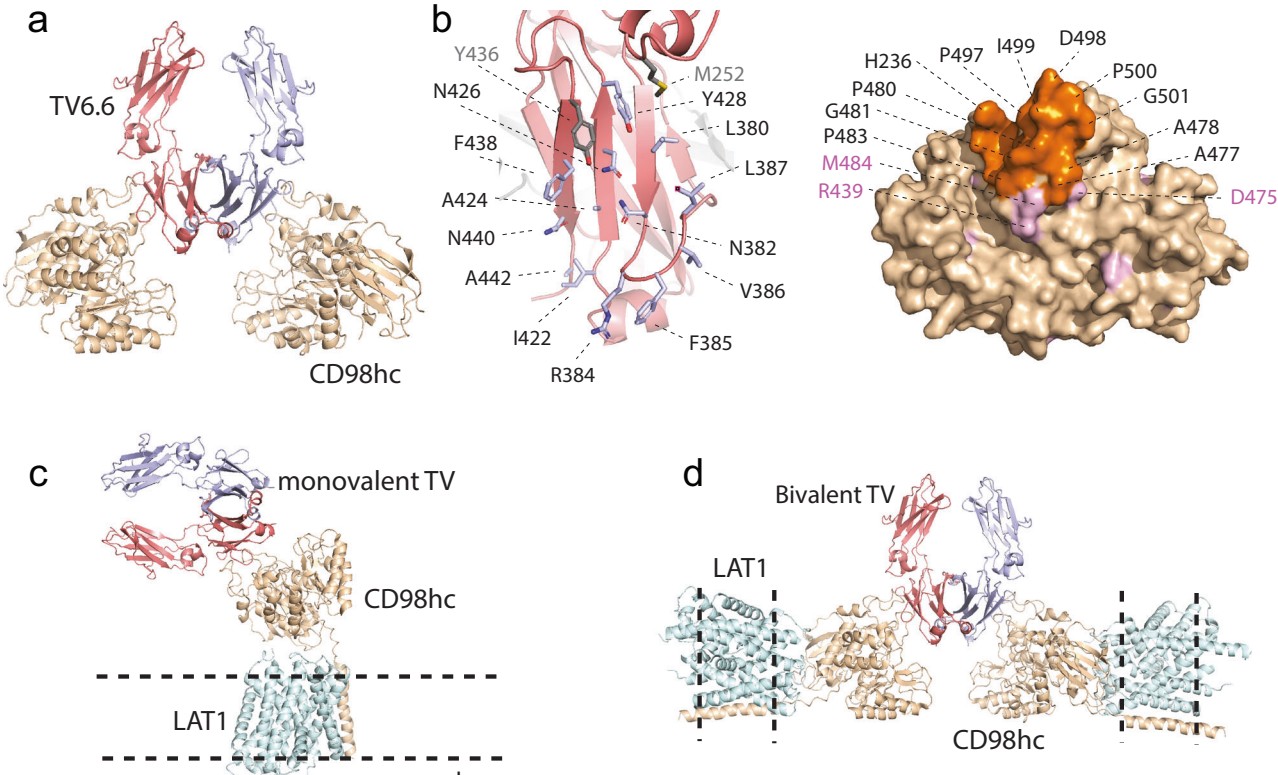

**Fig. 2 | Crystal structure of the TV6.6 and human CD98hc co-complex provides molecular details of the interaction. a** Model of Fc with TV6.6 (red and blue) in complex with two copies of CD98hc (tan); PDB ID: 8G0M **b** Zoom in of the TV6.6 surface with engineered residues (blue sticks) and wild-type positions (gray sticks) that contact CD98hc. Surface model of CD98hc indicating residues in contact with the TV (orange) and surface exposed positions that differ between human and cynomolgus (pink). **c, d** Model of monovalent (**c**) and bivalent (**d**) TV binding to the CD98hc-LAT1 complex (PDB ID: 6JMQ) on the cell surface.

enable further characterization (Fig. 1a)[47]. A representative monovalent CD98hc binding ATV clone bound HeLa cells expressing endogenous human CD98hc and rodent CHO cells over-expressing cynomolgus CD98hc, but not parental CHO cells, confirming specific binding to native human and cynomolgus CD98hc on the cell surface without general non-specific binding to cells (Fig. 1e). To further show that the TV mutations did not disrupt FcRn binding or introduce non-specific interactions, the plasma pharmacokinetics (PK) of ATV$^{CD98hc}$ variants, which do not bind mouse CD98hc, were assessed in wild type mice. After a single intravenous dose, ATV$^{CD98hc}$ variants were found to exhibit clearance parameters comparable to control IgG (Supplementary Table 1).

## Structural characterization of the TV6.6-CD98hc interaction

To gain insight into the molecular basis for the TV-CD98hc interaction, TV6.6 was co-crystallized with the extracellular domain of human CD98hc. The complex structure was refined to 2.25 Å resolution, with electron density of interacting residue sidechains well resolved (Fig. 2a, Supplementary Fig. 3a, and Supplementary Table 2). All engineered TV residues participate in binding to a structured loop region on CD98hc that lies adjacent to the α/β barrel structure, except for TV$^{Ala442}$, which was mutated to prevent the incorporation of a glycosylation motif (Fig. 2b). Surface complementarity at the complex interface is high, with a total buried surface area of 1551 Å$^2$, consistent with the 230 nM affinity of TV6.6 for human CD98hc (Fig. 1c). Structural determinants at the complex interface also agree with the engineered TV binding data. In particular, the TV makes van der Waals contacts with Glu229, Gln231, His236, Ala477-Pro483 Ala486, Val488, and Pro497-Pro500 of CD98hc, rationalizing the engineering of an extensive hydrophobic surface on the TV that includes the Glu380Leu,

Gly385Phe, Gln386Val, Pro387Leu, Val422Ile, Met428Tyr and Gln438Phe mutations (Figs. 1c, 2b, Supplementary Fig. 3a).

In the complex structure, the backbone of the β-sheet region on TV6.6 is minimally perturbed compared to wild-type Fc, with a Cα RMSD of 0.415 Å (Supplementary Fig. 3c). This is consistent with the limited functional sequence space within the TV6 family (Fig. 1c), as only a subset of amino acids satisfies the strict secondary structure requirements while preserving binding to CD98hc. Whereas a rigid backbone Fc structure is maintained, flexibility within the Pro480-Pro483 and Ser495-Ser504 loop regions on CD98hc likely contributes to high-affinity binding. This is reflected in a 3-4 Å shift for the latter, compared to canonical CD98hc structures (Supplementary Fig. 3b). All CD98hc residues interacting with the TV segment on TV6.6 are conserved between the human and cynomolgus CD98hc homologues; however, non-conserved residues proximal to the TV interface are expected to slightly alter the cynomolgus CD98hc structure (Fig. 2b). These observations likely explain the cross-reactive binding of TVs, which display a 3-10-fold decrease in affinity to cynomolgus CD98hc (Fig. 1d).

TV engagement at the cell surface was modeled by overlaying the TV6.6-CD98hc structure (PDB ID:8G0M) with an existing CD98hc/LAT1 complex structure (PDB ID:6JMQ; Fig. 2c, d). These structural models illustrate that both monovalent (Fig. 2c) and bivalent (Fig. 2d) TVs can bind CD98hc on the cell surface. Furthermore, they suggest a single bivalent TV can simultaneously engage two CD98hc/LAT1 complexes, although at an extreme angle that would seem to require considerable membrane curvature or further structural flexibility at the TV-CD98hc interface. As supported by the structure, bivalent ATV$^{CD98hc}$:DNP showed enhanced cell binding compared to its monovalent counterpart (Supplementary Fig. 3d).

## Generation of a humanized CD98hc mouse model

The TV6 family is not cross-reactive with mouse CD98hc, likely due to low sequence conservation between the extracellular domain of mouse and human CD98hc (71.9% identity, Supplementary Fig. 4). To enable in vivo characterization, a knock-in mouse model expressing chimeric CD98hc from the endogenous mouse *Slc3a2* (CD98hc) locus was generated (CD98hc[mu/hu] KI mice; Supplementary Fig. 5a and Supplementary methods). Expression patterns of CD98hc[mu/hu] were comparable to endogenous mouse CD98hc expression in the brain, kidney, pancreas, testis, and intestine, as well as a lack of expression in lung and liver (Supplementary Fig. 5b–h). Consistent with immunohistochemistry (IHC) data, Western blots confirmed CD98hc protein expression levels were similar across WT and homozygous CD98hc[mu/hu] KI mice (Supplementary Fig. 5i, j). These data suggest that CD98hc[mu/hu] KI mice faithfully recapitulate endogenous mouse CD98hc expression patterns and protein levels.

## Enhanced brain uptake of ATV[CD98hc] in humanized mice

To characterize exposure and safety of the TV[CD98hc] platform, monovalent and bivalent variants of ATV[CD98hc.6.8]:DNP were generated with and without mutations that attenuate binding to FcγRs (EF+ and EF-, Fig. 3a). After a single dose, ATV[CD98hc] variants had higher clearance from plasma compared to an isotype-matched control huIgG (Fig. 3b), consistent with CD98hc-mediated disposition in the periphery. Bivalent binding to CD98hc increased ATV[CD98hc] clearance, likely due to stronger apparent CD98hc affinity, while binding to FcγRs did not impact plasma exposure. Brain concentrations of ATV[CD98hc] variants increased until 7 days post-dose ($T_{max}$), at which time, levels were 6-8 fold higher than control huIgG. Concentrations of all variants remained significantly elevated (7–12 fold) at 21 days post-dose (Fig. 3c). Compared to monoATV[CD98hc], biATV[CD98hc] variants cleared slower from the brain, with concentrations of biATV[CD98hc] persisting in the brain two weeks after plasma concentrations were below the lower limit of quantification. Attenuation of FcγR binding marginally reduced brain exposure of ATV[CD98hc] (Fig. 3c).

Vascular and parenchymal brain cells were fractionated using the capillary depletion method[12,45,48] to evaluate whether the increase in total brain exposure was consistent with transcytosis of ATV[CD98hc] into the brain parenchyma (Supplementary Fig. 6a, b). Significantly elevated parenchymal concentrations of ATV[CD98hc] were observed at all tested timepoints, reaching 20–25 fold over control huIgG at 14 days post-dose (Fig. 3d). Increased ATV[CD98hc] levels were also observed in the vascular fraction (3–7 fold over control, Fig. 3e). These data demonstrate that ATV[CD98hc] has enhanced brain uptake relative to control huIgG resulting in sustained brain exposure after a single systemic dose. Furthermore, binding valency influences peripheral clearance, brain exposure, and kinetics. Additional time course studies were conducted to evaluate peripheral tissue uptake of mono and biATV[CD98hc] and revealed enhanced uptake in tissues known to express CD98hc (Supplementary Fig. 7a–p; Supplementary Fig. 5). No impact on CD98hc protein expression was observed in kidney and testes, where some of the highest CD98hc-mediated uptake was observed (Supplementary Fig. 7q–t).

To better understand how valency and FcγR engagement impact ATV[CD98hc] distribution within the brain, during the capillary depletion process, concentrations of ATV[CD98hc] in the cell-associated fraction (including both vascular and parenchymal cells) compared to the non-cell-associated supernatant fractions were examined (Supplementary Fig. 6c, d). Following homogenization of brain tissue, cells were centrifuged, and the cell-associated fraction was taken from the cell pellet (used to generate the vascular and parenchymal fractions) and the non-cell associated fraction was taken from the supernatant (see Materials and Methods for details). At 7 days post-dose ($T_{max}$), biATV[CD98hc] were observed to be more cell-associated compared to monoATV[CD98hc], while binding to FcγRs had no impact on cell

association (Fig. 3f). These data are consistent with the increased cell binding observed in vitro with biATV[CD98hc] (Supplementary Fig. 3d), presumably due to avid binding.

It has been previously demonstrated that the efficiency of brain delivery with TfR-binding BBB platforms is dependent on binding affinity and valency[26,32]. To investigate this relationship for CD98hc-binding molecules, brain exposures for mono- and biATV[CD98hc]:DNP affinity variants were measured 7 days ($T_{max}$) after a single systemic dose (Supplementary Fig. 6e, g). Across a 30-fold range (20–550 nM), affinity did not substantially impact brain exposure for either monovalent or bivalent ATV[CD98hc] variants, although uptake diminished as affinity was further reduced for biATV[CD98hc] (2100 nM; Supplementary Fig. 6e, g). Consistent with the correlation between the increased apparent affinity of biATV[CD98hc] molecules and an increase in cell association (Fig. 3f, Supplementary 3d), clones having stronger affinity for CD98hc were observed to be more cell associated (Supplementary Fig. 6f, h).

## Valency and FcγR binding impact CNS cell type biodistribution of ATV[CD98hc]

To gain insight into the biodistribution of ATV[CD98hc] molecules, the localization of each ATV[CD98hc] variant was assessed by IHC. Low magnification sagittal sections revealed broad distribution of ATV[CD98hc] molecules throughout brain regions (Supplementary Fig. 8). At higher magnification, evaluation of ATV[CD98hc] distribution over time showed more prominent vascular staining of ATV[CD98hc] at 1-day post-dose, which transitioned to a more diffuse parenchymal staining by 7 days post-dose that persisted for the duration of the study (Fig. 3g). Diffuse punctate staining was observed throughout the parenchyma for all variants except for monoATV[CD98hc]:DNP EF+, which exhibited a distinct cellular biodistribution pattern. To further characterize these patterns, huIgG immunoreactivity was examined in sections co-stained with multiple cell-specific markers. All ATV[CD98hc] variants colocalized to some degree with AQP4, an integral membrane protein predominantly expressed on astrocyte endfeet and, to a lesser extent, other astrocyte processes[49,50] (Fig. 4a). This is consistent with CD98hc expression on both endothelial cells and perivascular astrocyte endfeet, as suggested by CD98hc colocalization with both Glut1, a brain endothelial cell marker, and AQP4, respectively (Supplementary Fig. 9a, c, d). Of all ATV[CD98hc] variants tested, only monoATV[CD98hc]:DNP EF+ colocalized strongly with Iba1 (Fig. 4b). Very little endogenous CD98hc expression was seen on Iba1-positive microglia (Supplementary Fig. 9b), suggesting this microglia localization is driven by FcγR binding and is sensitive to TV valency. Colocalization of huIgG was not observed with the neuronal marker NeuN (Supplementary Fig. 10). Taken together, these data indicate that ATV[CD98hc] cell-specific biodistribution in the CNS is non-neuronal, restricted to subsets of glial cells, and can be influenced by both valency and FcγR binding.

## ATV[CD98hc] and ATV[TfR] have distinct kinetics

ATV[CD98hc] shows remarkably slower uptake, more prolonged brain exposure, and a distinct brain biodistribution pattern compared to what has been previously reported for TfR binding platforms[12,26,51]. To directly compare ATV platforms that bind these two targets in vivo, a double KI mouse expressing human/mouse chimeric versions of both CD98hc and TfR was generated (CD98hc[mu/hu], TfR[mu/hu] KI). ATV[CD98hc.6.8] (170 nM $K_D$) or ATV[TfR.35.23.4] (620 nM $K_D$), both monovalent and EF- with DNP Fabs, were systemically administered in CD98hc[mu/hu], TfR[mu/hu] KI mice. These variants were chosen as these affinities to TfR and CD98hc have been shown to provide high brain exposures (Supplementary Fig. 6e) while balancing peripheral target-mediated disposition[12]. ATV[TfR] exhibited higher plasma clearance compared to ATV[CD98hc] (Fig. 5a). Consistent with previous reports, TfR binding enabled rapid brain uptake with a $T_{max}$ at 1 day post-dose, whereas brain concentrations of ATV[CD98hc] gradually increased until $T_{max}$ was reached

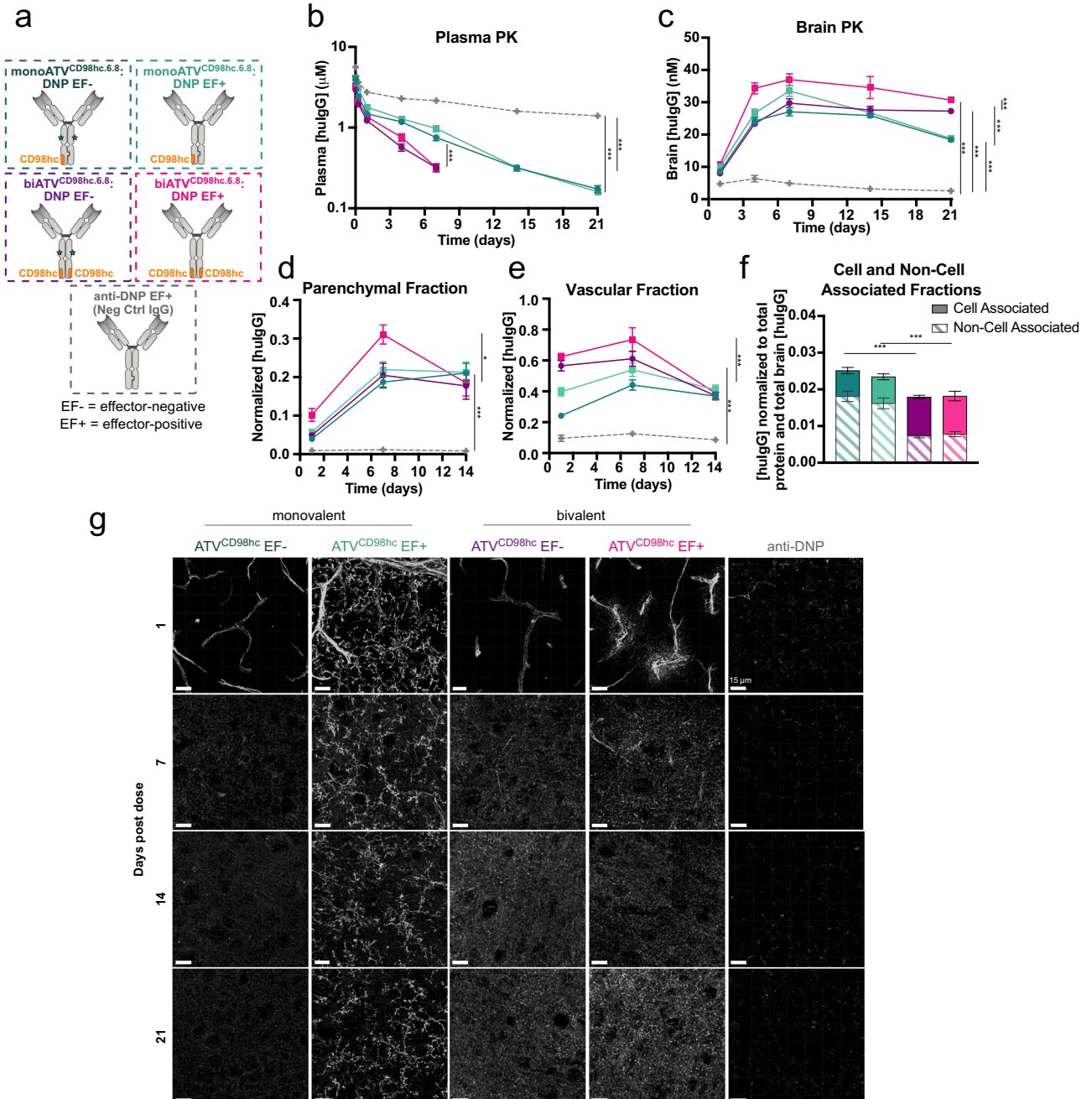

**Fig. 3 | Monovalent and bivalent ATV^CD98hc6.8.:DNP have prolonged brain exposure after a single dose in CD98^mu/hu KI mice. a** Cartoons of monovalent and bivalent ATV^CD98hc6.8.:DNP, with (EF−) and without (EF+) mutations to mitigate FcγR binding. **b−c** Brain and plasma concentrations of ATV^CD98hc6.8.:DNP variants (170 nM $K_D$) after a single intravenous (IV) 50 mg/kg dose in CD98hc^mu/hu KI mice. Plasma clearance values were 19.6−21.9 mL/d/kg and 43−49 mL/d/kg for monoATV^CD98hc6.8.:DNP and biATV^CD98hc6.8.:DNP, respectively, compared to 7.1 and 8.3 mL/d/kg for mono- or biATV^CD98hc6.8.:DNP in WT mice (Supplementary Table 1). All ATV^CD98hc variants had higher plasma clearance and brain exposure compared to the anti-DNP control, and biATV^CD98hc6.8.:DNP variants had higher plasma clearance and brain exposure compared monoATV^CD98hc6.8.:DNP variants. FcγR binding did not impact plasma clearance and was associated with a small increase in brain exposure. **d−e** Concentrations of huIgG in brain parenchymal (**d**),

vascular fractions (**e**), and cell-associated and non-cell associated fractions (**f**) obtained by capillary depletion at 7 days post dose. There were increased concentrations of all ATV^CD98hc TV variants in the parenchymal and vascular fractions. Consistent with whole-brain exposure, biATV^CD98hc6.8.:DNP EF+ had the highest concentration in the parenchyma and biATV^CD98hc6.8.:DNP variants were found at higher concentrations in the vascular fraction. biATV^CD98hc6.8.:DNP variants were also more cell associated than monoATV^CD98hc6.8.:DNP variants. **b−e** Two-way ANOVA. **f** One-way ANOVA. **b−f** Graphs display mean ± SEM, $n = 4–5$, see Source Data for exact n/group. $*p < 0.05$, $****p < 0.0001$. See Supplementary Table 6 for exact $p$ values. **g** Immunohistochemical localization of ATV^CD98hc6.8.:DNP variants in brain cortical sections as assessed with anti-huIgG across timepoints. Scale bars = 15 μm. Representative image shown from $n = 5$/group, $n = 2$ stained sections per animal. Source data are provided as a Source Data file.

5−7 days post-dose (Fig. 5b). Once in the brain, elevated levels of ATV^CD98hc persisted for the duration of the study (21 days), whereas ATV^TfR concentrations diminished and were undetectable after 7 days (Fig. 5b). Capillary depletion data confirmed both ATVs crossed the

BBB into the parenchymal brain fraction (Supplementary Fig. 11a−c). These data demonstrate that binding to either CD98hc or TfR can enable enhanced brain uptake compared to control huIgG, although each has a distinct kinetic profile.

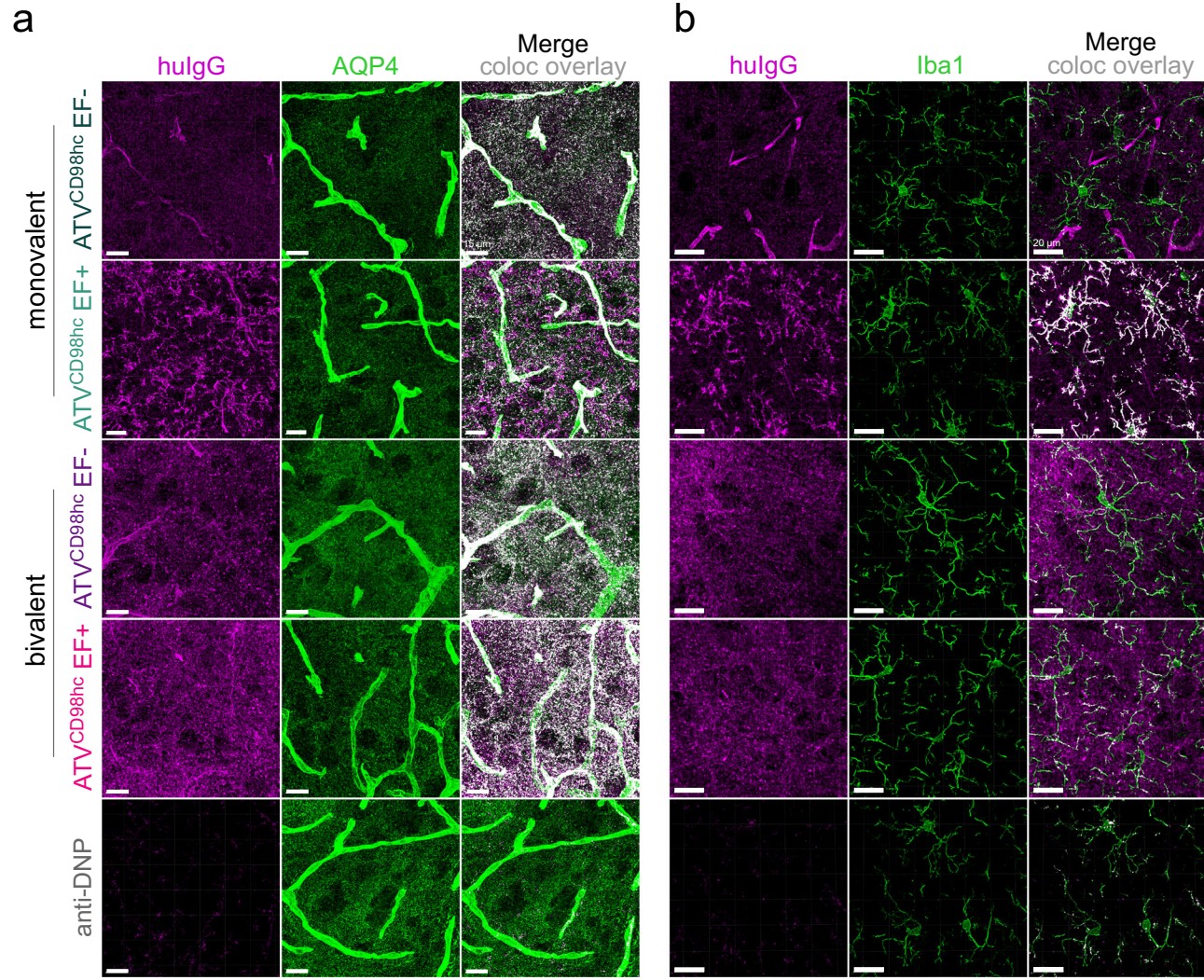

**Fig. 4 | Biodistribution of ATV$^{CD98hc6.8}$:DNP variants in the brain is dependent on valency and FcγR binding in CD98hc$^{mu/hu}$ KI mice. a** Immunohistochemical cell-specific localization of ATV$^{CD98hc6.8}$:DNP variants (170 nM $K_D$) in brain cortical sections with antibodies against huIgG (purple) and AQP4 (**a**, green, scale bars = 15 μm) or Iba1 (**b**, green, scale bars = 20 μm). Overlays are shown with colocalization pseudocolored in white. Representative image shown from n = 5/group, n = 2 stained sections per animal.

The trafficking and retention of ATV$^{CD98hc}$ and ATV$^{TfR}$ were compared using HEK293 cells that endogenously express both TfR and CD98hc to explore potential mechanisms driving these distinct in vivo behaviors at the cellular level. Immunocytochemistry staining of total huIgG under permeabilizing conditions revealed substantially more intracellular ATV$^{TfR}$:DNP compared to ATV$^{CD98hc}$:DNP (Supplementary Fig. 12b, e). Conversely, non-permeabilizing staining revealed more persistent cell surface localization over time for ATV$^{CD98hc}$:DNP compared to ATV$^{TfR}$:DNP (Supplementary Fig. 12a, d, f). Additionally, there was greater cellular retention with ATV$^{CD98hc}$:DNP compared to ATV$^{TfR}$:DNP, as measured by huIgG in the cell lysate after treatment washout (Supplementary Fig. 12c). While the exact rate of internalization may vary in different cell types, these in vitro data support the in vivo observation that ATV$^{TfR}$:DNP is more rapidly internalized and degraded compared to ATV$^{CD98hc}$:DNP. It also provides a cellular trafficking mechanism by which ATV$^{CD98hc}$ and ATV$^{TfR}$ differentiate in their exposure and kinetic profiles in vivo.

## ATV$^{CD98hc}$ kinetics can be modulated with targeted Fabs

The observation that biATV$^{CD98hc}$ persists in brain long after plasma concentrations are undetectable (Fig. 3b, c) suggests that retention, in addition to uptake, may play a significant role in the brain exposure

profiles of CD98hc-binding ATVs. To evaluate whether CD98hc-mediated distribution and retention could be modulated by additional binding, Fabs targeting and inhibiting the activity of β-secretase 1 (BACE1) were fused to mono- and biATV$^{CD98hc.6.39}$ (94 nM $K_D$ to CD98hc). BACE1 is an amyloid precursor protein cleavage enzyme, and antibodies targeting BACE1 primarily localize to neuronal endosomes and inhibit the production of endogenous Aβ[52,53]. The addition of BACE1 binding to ATV$^{CD98hc}$ may therefore potentially impact cell specificity, trafficking, and retention in brain of ATV$^{CD98hc}$. ATV$^{CD98hc}$:BACE1 variants were administered intravenously to CD98hc$^{mu/hu}$ KI mice, and plasma and brain PK were evaluated out to 21 days. The resulting maximum brain concentrations for ATV$^{CD98hc}$:BACE1 variants were ~50% lower than the concentrations observed with ATV$^{CD98hc}$:DNP (Fig. 5d compared to Fig. 5b and Fig. 3c). In contrast to the persistent brain exposure of ATV$^{CD98hc}$:DNP, ATV$^{CD98hc}$:BACE1 concentrations declined by 14 days to levels similar to control anti-BACE1 (Fig. 5d). Plasma clearance for monovalent and bivalent ATV$^{CD98hc}$:BACE1 and ATV$^{CD98hc}$:DNP were similar (Fig. 5c compared to Fig. 5a and Fig. 3b), indicating BACE1 binding had minimal impact on peripheral clearance, and that differences in brain exposures were not simply a consequence of altered systemic exposures. To confirm parenchymal delivery of ATV$^{CD98hc}$:BACE1 and inhibition of BACE1 activity, endogenous Aβ40

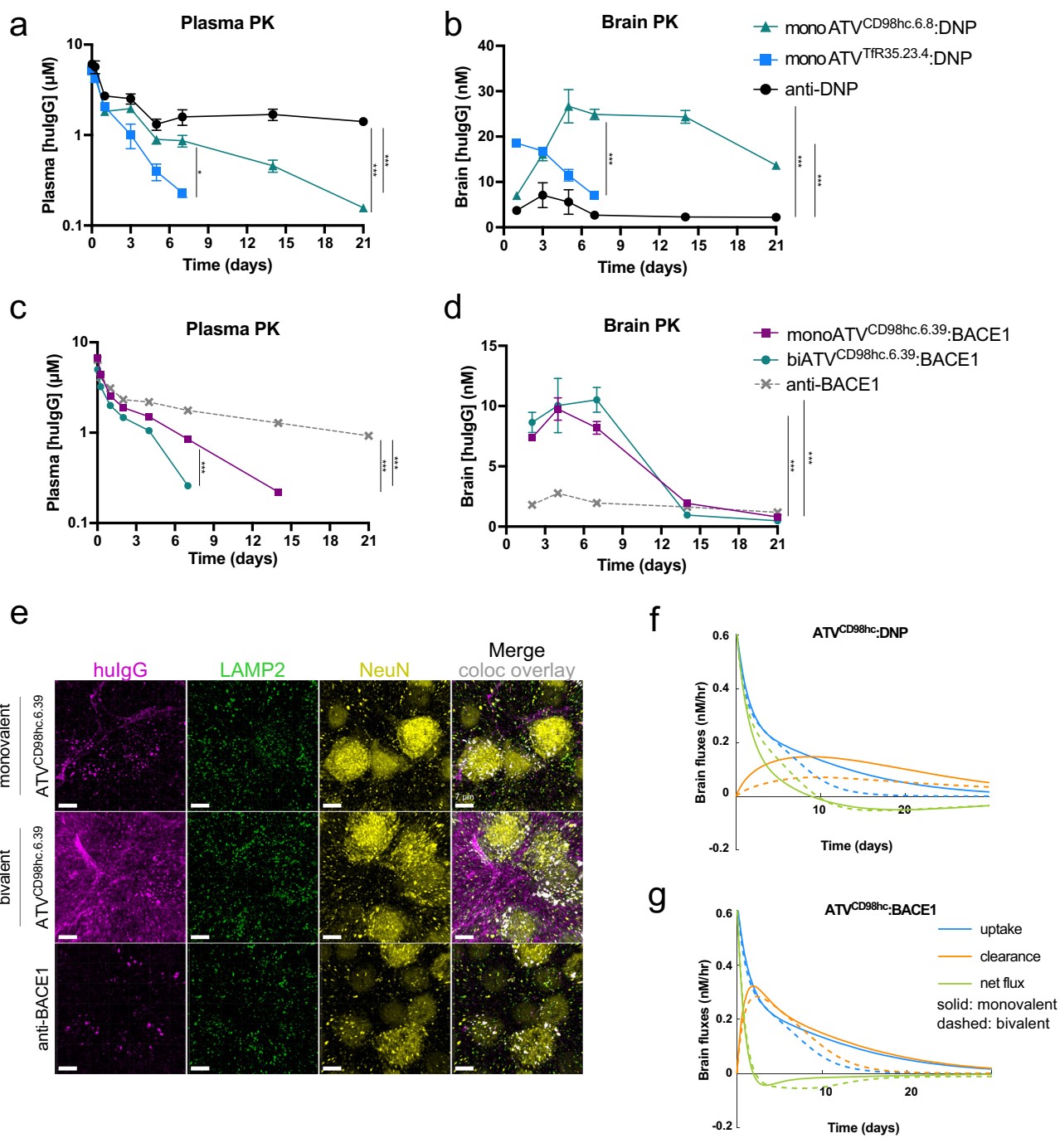

**Fig. 5 | Pharmacokinetics and distribution of ATV$^{CD98hc}$ are distinct from ATV$^{TfR}$.**
**a**, **b** PK of ATV$^{CD98hc.6.8}$:DNP (170 nM $K_D$) and ATV$^{TfR.35.23.4}$:DNP (620 nM $K_D$) in plasma (**a**) and brain (**b**) following a single 50 mg/kg IV dose. Plasma clearances for both ATVs were increased relative to anti-DNP (5.3 mL/d/kg), and plasma clearance for ATV$^{TfR.35.23.4}$:DNP (39.0 mL/d/kg) was increased compared to ATV$^{CD98hc.6.8}$:DNP (17.7 mL/d/kg). Both ATV$^{CD98hc.6.8}$:DNP and ATV$^{TfR.35.23.4}$:DNP had increased brain exposure compared to control and had different exposure profiles compared to each other. **c**, **d** PK of mono- and biATV$^{CD98hc.6.39}$:BACE1 (94 nM $K_D$) in plasma (**c**) and brain (**d**) following a single 50 mg/kg IV dose. Both mono- and biATV$^{CD98hc.6.39}$:BACE1 had higher plasma clearance compared to control, and biATV$^{CD98hc.6.39}$:BACE1 (35.9 mL/d/kg) had higher plasma clearance compared to monovalent (19.9 mL/d/kg). Compare to the plasma clearance measured for mono- and biATV$^{CD98hc.6.8}$:DNP (19.6 and 43.9 mL/d/kg, respectively Fig. 3b). Both mono- and biATV$^{CD98hc.6.39}$:BACE1

had increased brain exposure compared to control, and were not significantly different from each other. **a**–**d** Graphs display mean ± SEM, $n = 5$/group. Two-way ANOVA. Data points not shown were <LLOD. *$p < 0.05$, ****$p < 0.0001$. See Supplementary Table 6 for exact $p$ values. **e** Immunohistochemical localization of huIgG for mono- and biATV$^{CD98hc.6.8}$:BACE1 (purple) in cortical brain sections, LAMP2 (green), and NeuN (yellow) 7 days after a single 50 mg/kg IV dose. Scale bars = 7 μm. Overlays are shown with colocalization of huIgG and Lamp2 within NeuN positive neurons is pseudocolored in white. Representative images are shown from $n = 5$ animals/treatment group, $n = 2$ IHC sections per animal. **f**, **g** Simulated brain uptake, clearance, and total net flux for mono- and biATV$^{CD98hc}$:DNP (**f**) and ATV$^{CD98hc}$: BACE1 (**g**). Simulations are performed for a 50 mg/kg IV dose of ATV$^{CD98hc}$, CD98hc $K_D = 170$ nM. Source data are provided as a Source Data file.

levels in brain were measured and found to be significantly reduced at all timepoints when concentrations of ATV[CD98hc]:BACE1 were greater than control huIgG (Supplementary Fig. 11d). These data suggest that a lack of baseline localization to neurons does not preclude the ability of ATV[CD98hc] to bind to, and impact the function of, neuronal Fab targets.

To better understand how BACE1 binding alters exposure of ATV[CD98hc], biodistribution was assessed by IHC on cortical brain sections after dosing. In contrast to the lack of neuronal colocalization observed with ATV[CD98hc]:DNP (Supplementary Fig. 10), internalization of mono- and biATV[CD98hc]:BACE1 molecules into neurons was observed (Fig. 5e and Supplementary Fig. 11e). Furthermore, whereas monoATV[CD98hc]:BACE1 localized predominantly to LAMP2-positive neuronal lysosomes, biATV[CD98hc]:BACE1 distributed to both neuronal lysosomes and in non-neuronal puncta consistent with putative astrocyte process localization (Fig. 5e). This valency-dependent distribution also suggests that avid binding to CD98hc can compete with BACE1 Fab engagement to preserve some CD98hc-dependent localization. Taken together, these data suggest that CD98hc-mediated retention can be modulated by other aspects of the molecule (e.g., Fab binding), leading to altered cellular localization, trafficking, and clearance of ATV[CD98hc].

## Analysis of brain PK of ATV[CD98hc] molecules using mathematical modeling

Insights from the ATV[CD98hc]:BACE1 experiment suggest that ATV[CD98hc] retention in brain, in addition to uptake, drives the prolonged brain exposures observed in the absence of Fab binding. To strengthen this hypothesis, a multi-compartment pharmacokinetic model of ATV[CD98hc] uptake and clearance processes was developed. The model was calibrated to available systemic and brain ATV[CD98hc] PK data (Supplementary materials and methods) and was used to conduct a mathematical analysis to better understand the dynamics underlying the observed brain PK of ATV[CD98hc] molecules (Supplementary Fig. 13a). Calibrations were performed for mono- and biATV[CD98hc] over a wide range of CD98hc affinities with and without target binding Fabs, with model results showing good agreement with the collective observed experimental data, which was largely collected on ATV[CD98hc] EF+ variants (Supplementary Fig. 13b–h). Notably, in the absence of Fab binding, model analysis suggests a reduced clearance of ATV[CD98hc] from brain compared to that of control huIgG (Supplementary Fig. 13h, Supplementary Tables 3, 4). The modeling results are also consistent with brain clearance exhibiting a dependency on both CD98hc affinity and binding valency, with lower clearance predicted for clones with tighter CD98hc binding (Supplementary Fig. 13h).

Simulations of the temporal dynamics of ATV[CD98hc] molecules entering (i.e., uptake) and leaving (i.e., clearance) the brain were also performed to further characterize how the interplay between these fluxes give rise to the experimentally observed brain PK (Fig. 5f, g). The same analysis was done to model the dynamics of a standard anti-DNP antibody (Supplementary Fig. 13i). Across simulations of both mono- and biATV[CD98hc]:DNP for a CD98hc affinity of 170 nM (chosen to match molecules for which the PK is shown Fig. 3b, c), elevated systemic exposures drive brain uptake and the net flux (i.e., the difference between the uptake and clearance) remains positive before $T_{max}$. A protracted falling phase occurs after $T_{max}$ as uptake falls below clearance of ATV[CD98hc] from brain. During this latter phase, CD98hc-mediated retention offsets the diminished uptake, leading to a slow decay in brain concentrations. Model simulations suggest biATV[CD98hc] has increased net flux prior to $T_{max}$, which is driven largely by reduced brain clearance and results in higher brain $C_{max}$ compared to their monovalent counterparts. Additionally, the increased retention of biATV[CD98hc] due to a higher apparent affinity allowed elevated brain concentrations to persist well after uptake returns to zero (Figs. 3c, 5f). These model-informed conclusions are in line with mouse PK data and provide a functional explanation for the observation that both

stronger affinity and bivalent binding led to increased cell association within the brain (Fig. 3f, Supplementary Fig. 6f, h).

The effect of CD98hc-mediated retention on brain exposure was further explored by comparing simulations of the fluxes for ATV[CD98hc]:DNP and ATV[CD98hc]:BACE1 at the same affinity (Fig. 5f, g). While brain uptake should not differ between these two formats, simulated clearance of ATV[CD98hc]:BACE1 was markedly higher compared to ATV[CD98hc]:DNP as BACE1 binding drives neuronal internalization and lysosomal degradation, resulting in a pronounced reduction in both duration and maximum of elevated brain concentrations (Fig. 5f, g). Collectively, this model offers a more detailed explanation for the intricacies of the PK profiles observed in mouse, and further supports the conclusion that CD98hc-mediated retention, in addition to uptake, is a determining factor in the measured brain concentrations of ATV[CD98hc].

## Repeat dosing of ATV[CD98hc] in mice yielded no observed safety findings

Translation of the TV[CD98hc] platform requires understanding the safety profile under repeat dosing conditions and demonstration of maintained transport capacity over time. To address this, ATV[CD98hc]:DNP variants were administered weekly to CD98hc[mu/hu] KI mice for one month. Plasma exposures at $C_{max}$ and $C_{trough}$ remained consistent across the duration of the study and with what was previously observed in single-dose studies (Fig. 6a, c). Brain concentrations of both mono- and biATV[CD98hc] 24 h after the last dose were higher than after a single dose (Fig. 6b,d compared to Figs. 3c, 5b), demonstrating accumulation in brain as a function of elevated plasma exposures driving uptake as well as CD98hc-mediated retention in brain. Hematology parameters were within normal limits for all treatment groups. Specifically, there was no impact on CD98hc-expressing monocytes or lymphocytes (Fig. 6e-f, h-i)[41,54]. No impact on reticulocytes was observed with ATV[CD98hc]:DNP variants (Fig. 6g, j), in contrast to what has been previously reported for effector positive and bivalent TfR binding molecules[51]. Furthermore, no significant findings attributable to the administration of ATV[CD98hc]:DNP were present by histopathological evaluation of tissues with high CD98hc expression (i.e., kidney, pancreas, testis, intestines, skin, and brain). Taken together, these data suggest CD98hc brain transport capacity is maintained after multi-dosing and is well tolerated in mice.

## CD98hc-dependent brain uptake and biodistribution translates to cynomolgus monkey

Investigating the translation of TV[CD98hc] to a non-human primate species enables better predictions of biodistribution in humans and is a necessary step towards the clinical application of the platform. To that end, ATV[CD98hc.6.29]:DNP variants were evaluated in a small pilot study to qualitatively compare ATV[CD98hc] biodistribution and brain uptake in cynomolgus monkey with what was observed in mouse. Non-targeting Fabs were used for this initial characterization of baseline platform performance as targeting Fabs, such as anti-BACE1, were observed to significantly influence brain uptake, clearance, and localization within the CNS in mouse (Fig. 5b, d).

Following a single peripheral dose, higher plasma clearance was observed for biATV[CD98hc] relative to control antibody (Fig. 7a). Brain uptake of all ATV[CD98hc] variants was seen in both whole brain and the parenchymal fraction after capillary depletion (Fig. 7b, Supplementary Fig. 14a, b, e), confirming transcytosis of ATV[CD98hc] across the BBB and into the parenchyma. Despite lower peripheral exposure, biATV[CD98hc] resulted in higher brain and vascular concentrations compared to monoATV[CD98hc] (Fig. 7a, b, Supplementary Fig. 14f). Non-cell associated monoATV[CD98hc] molecules contributed notably to whole brain huIgG concentrations, whereas biATV[CD98hc] molecules were more cell associated (Supplementary Fig. 14c, d, g), consistent with bivalent binding

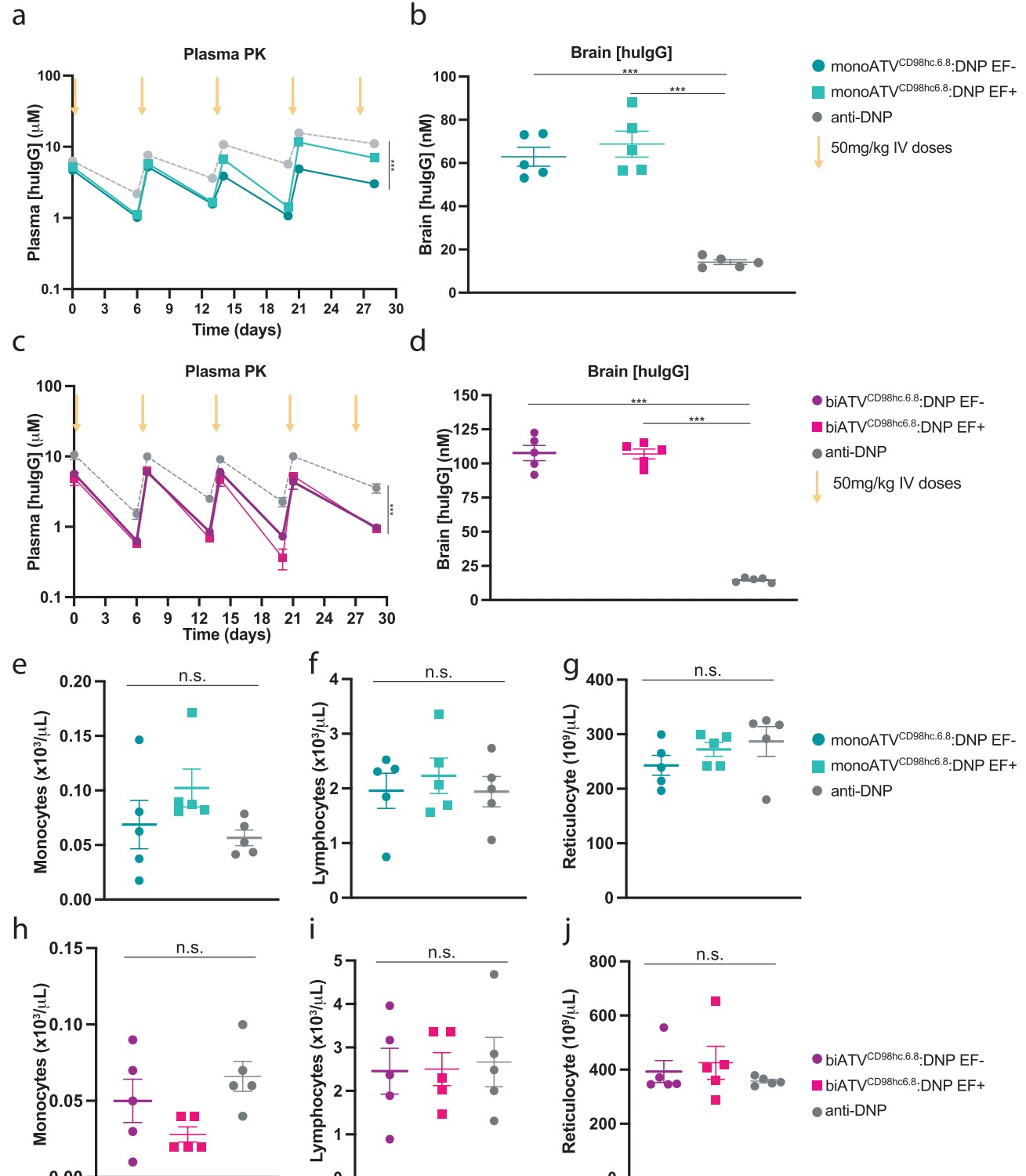

**Fig. 6 | Multidose of ATV^CD98hc6.8:DNP variants in CD98hc^mu/hu KI mice accumulate in brain without impacting circulating cells. a, c** Plasma PK of mono- and biATV^CD98hc6.8:DNP (170 nM $K_D$) with (EF−) and without (EF+) mutations to mitigate FcγR binding after 5 weekly IV doses (50 mg/kg). Plasma concentrations were assessed at 30 min after each dose ($C_{max}$) and 24 h before the next dose ($C_{trough}$). Two-way ANOVA. **b, d** Concentrations of ATV^CD98hc:DNP variants in brain 24 h after the final dose. One-way ANOVA ****$p < 0.0001$. See Supplementary Table 6 for exact $p$ values. **e–j** Circulating cell counts for monocytes (**e**, **h**), lymphocytes (**f**, **i**) and reticulocytes (**g**, **j**) as measured by complete blood count 24 h after the final dose. Cell numbers compared with a one-way ANOVA. **a–j** All graphs display $n = 4$–5/ group (see Source Data for exact n/group), mean ± SEM. Source data are provided as a Source Data file.

leading to improved CNS retention (Fig. 5f). FcγR binding had minimal impact on measured huIgG concentrations (Fig. 7a, b).

Cell type biodistribution of ATV^CD98hc in cynomolgus monkeys also closely resembled observations in mice and was consistent with

CD98hc expression in cynomolgus monkey brain (Supplementary Fig. 9e, f). MonoATV^CD98hc EF- and biATV^CD98hc EF+ colocalized with AQP4-positive astrocyte processes and endfeet (Fig. 7c), while both mono- and biATV^CD98hc EF+ colocalized with Iba1-positive microglia

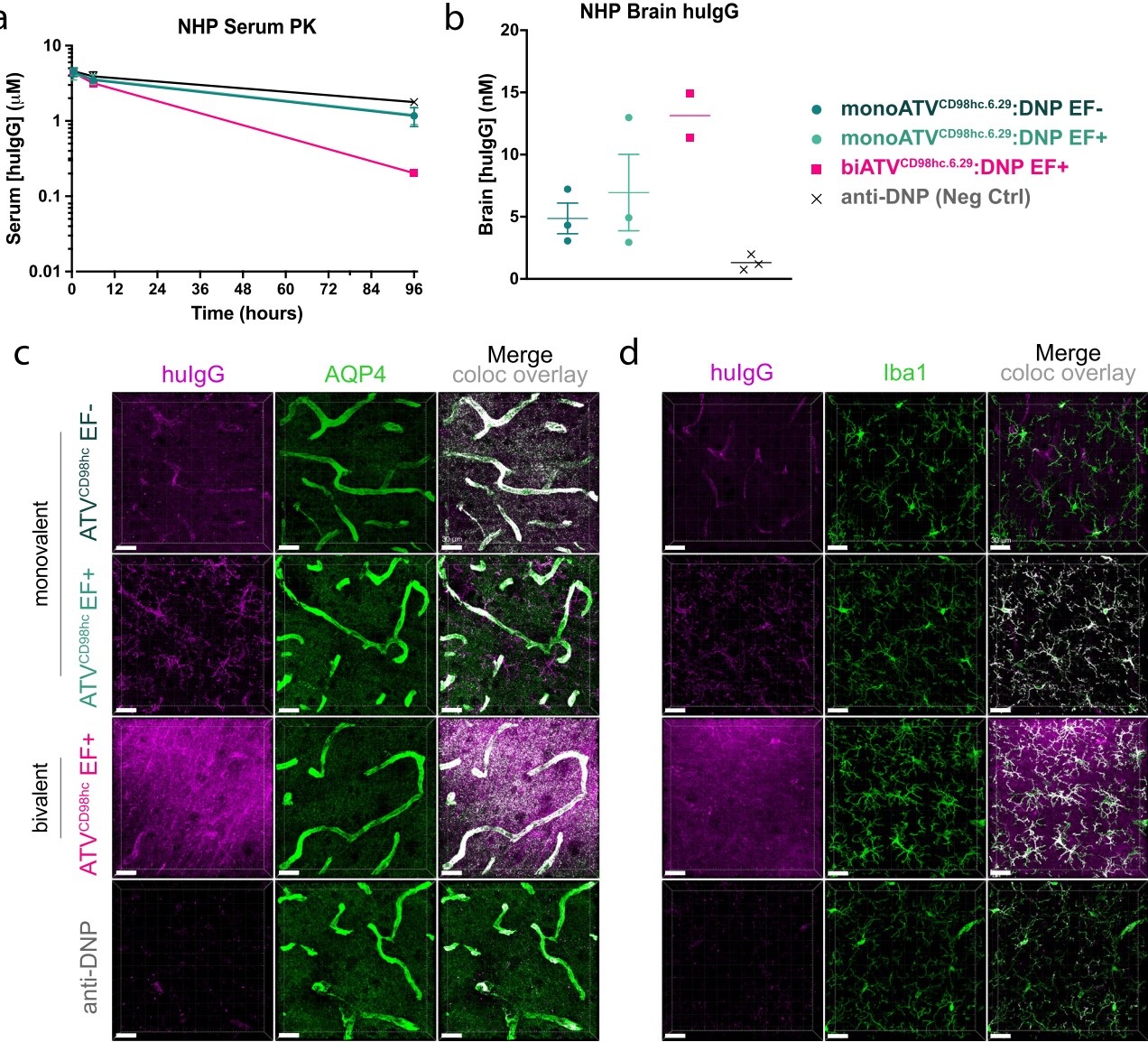

**Fig. 7 | ATV^CD98hc uptake is enhanced in brain and localize to astrocytes and microglia in cynomolgus monkeys. a, b** Plasma and brain exposure of mono- and biATV^CD98hc6.29:DNP (205 nM $K_D$ to cyno CD98hc) EF+ and monoATV^CD98hc6.29:DNP EF- in cynomolgus monkeys 4 days after a single 30 mg/kg IV dose. Graphs display mean ± SEM (for n = 2 or 3 as noted in figure) **c, d** Localization of ATV^CD98hc6.29:DNP variations in cortical brain sections assessed by IHC with antibodies against for

huIgG (purple) and AQP4 (**c**, green) or Iba1 (**d**, green). Scale bars = 30 μm. Overlays are shown with colocalization pseudocolored in white. Representative image shown from n = 3 animals/treatment group (except for biATV^CD98hc6.29:DNP, where one animal was excluded), n = 3 stained sections per animal. Source data are provided as a Source Data file.

(Fig. 7d, compare to Fig. 4a, b). None of the ATV^CD98hc variants tested showed localization to neurons (Supplementary Fig. 14h), consistent observations in mice.

In summary, our data suggests CD98hc binding TVs can successfully enhance brain uptake in cynomolgus monkeys, resulting in a biodistribution pattern similar to what was observed in our mouse studies. The findings across mice and cynomolgus monkeys, combined with proteomics data demonstrating high CD98hc protein expression on isolated human brain microvessels (Supplementary Fig. 15), collectively support further work on TV^CD98hc as a promising and differentiated BBB-crossing platform for neurotherapeutics.

## Discussion

The blood-brain barrier presents a unique challenge in drug development for neurodegenerative diseases, often restricting CNS drug concentrations to levels below those required for therapeutic benefit. We have previously described the engineering and preclinical

characterization of TV^TfR, an Fc-based human TfR-targeted drug delivery platform that has enabled antibody-, enzyme- and protein-based programs to advance towards clinical evaluation in Alzheimer's disease, neuronopathic lysosomal storage disease, and fronto-temporal degeneration[36,37]. Here, we have extended the TV approach by engineering and characterizing TV^CD98hc, a new TV platform that binds CD98hc. TV^CD98hc leverages a differentiated RMT target that enables prolonged brain exposures, compatibility with full effector function, and distinct and tunable brain cell type specificity in mice and monkeys.

Prior engineering of the Fc has focused on loop regions within the CH3 domain[12,55], leaving much of the sequence space unexplored. While engineering molecular recognition into β-sheets has been previously described[56–58], the present work extends β-sheet engineering to the Fc domain. By incorporating additional binding into a new region on the Fc, we further establish that much of the antibody scaffold can be modified to increase functionality. Furthermore, this

work illustrates the modularity and versatility of the TV approach for binding diverse RMT targets to enable brain delivery of biotherapeutics.

The lack of a humanized CD98hc mouse model has previously limited the characterization of human-specific CD98hc binding platforms[30]. Here, we develop a humanized CD98hc$^{mu/hu}$ KI mouse model and use it to show that ATV$^{CD98hc}$ in a monovalent and bivalent format have enhanced brain uptake relative to unmodified control antibodies. In addition, we show ATV$^{CD98hc}$ has lower peripheral clearance, prolonged brain exposure, and slower trafficking kinetics associated with brain uptake and clearance relative to ATV$^{TfR}$. While RMT platforms that utilize TfR are characterized by rapid internalization, CD98hc targeting allows for the possibility of brain delivery of biotherapeutics with relatively long exposure. Importantly, the consistency observed in the brain uptake and biodistribution of ATV$^{CD98hc}$:DNP between cynomolgus monkey and the humanized CD98hc$^{mu/hu}$ KI mouse supports continued use of the mouse as a representative model for further TV$^{CD98hc}$ characterization. However, it will be important to further confirm translatability as the TV$^{CD98hc}$ platform is used to enable more targeted biotherapeutics.

Given the dynamic nature of large molecule PK in the CNS, the interplay among mechanisms affecting the transport of drug into and out of the brain cannot be solely inferred by static measurements of concentrations. Mathematical modeling allowed us to gain a deeper understanding of the factors underlying the unique in vivo behavior of the ATV$^{CD98hc}$ platform. A key insight from the model is that CD98hc binding protects ATV$^{CD98hc}$ from clearance out of the CNS in an affinity and valency-dependent manner. This offers an explanation for how elevated brain concentrations are possible after active uptake has ceased. Our findings emphasize the risk of overinterpreting single static brain concentrations as a surrogate for dynamic measurements of brain uptake kinetics. Further work defining the uptake capacity of different RMT-based platforms is therefore necessary to understand how brain concentrations relate to overall brain exposure. Additionally, profiling brain uptake and distribution when affinities to CD98hc are significantly stronger than what has been achieved with the TV$^{CD98hc}$ platform will be a key area for further work. Diminishing brain uptake has been observed for TfR-binding molecules with strong affinities[26]. Establishing whether a similar relationship exists when targeting CD98hc will be important.

In addition to prolonged uptake, we show that ATV$^{CD98hc}$ with non-targeted Fabs displays baseline distribution in the brain to astrocyte processes and endfeet. This pattern is notably in contrast to the neuronal localization with ATV$^{TfR}$ and other TfR-based BBB platforms[12,26,32]. We also demonstrate how ATV$^{CD98hc}$ localization to astrocytes may be modified by the introduction of orthogonal binding to other targets. We show that the addition of anti-BACE1 Fab binding to a TV$^{CD98hc}$ drives localization to neurons, which express BACE1. More generally, this shift to neuronal localization demonstrates that Fab target binding can redirect ATV$^{CD98hc}$ to other cell types (e.g., neurons), even if the platform's base case neuronal localization is minimal. These shifts in biodistribution also impact the kinetics of brain exposure by driving intracellular localization and reducing the CD98hc-mediated retention, which may be desirable for certain therapeutic applications. Similarly, we show that ATV$^{CD98hc}$ with intact FcγR binding has altered biodistribution and localizes to microglia but does not impact the kinetics of brain exposure. Additionally, ATV$^{CD98hc}$ engagement with FcγR appears to be safe in mice even after multiple doses. This lack of observed safety findings is consistent with a phase 1 trial of an effector positive, bivalent anti-CD98hc antibody ($K_D$ to human CD98hc of 2 – 6 nM)[59]. Taken together, these observations suggest that ATV$^{CD98hc}$ can be directed by tuning the collective binding profile of the entire molecule.

The availability of the complementary TV$^{TfR}$ and TV$^{CD98hc}$ platforms provides an opportunity to tailor platform selection based on cell type-specific therapeutic targeting and desired mechanisms of action. TV$^{TfR}$ may be preferred for therapeutic targets that require cellular internalization, targeting within the endolysosomal pathway (e.g., enzyme replacement therapy for lysosomal storage diseases), or therapeutics that rely on achieving rapid $C_{max}$. TV$^{CD98hc}$ may be well suited for targets that are extracellular, on the cell surface requiring extended target engagement (e.g., antagonists), glial-specific, requiring FcγR engagement, or would benefit from sustained brain exposure. The distinct biodistribution, pharmacokinetics, and brain exposure enabled by TV$^{CD98hc}$ differentiates it from existing RMT-based approaches and may ultimately facilitate its potential application to tackle the new targets, novel biology, and different dosing regimens required for the treatment of the diverse array of CNS disorders.

## Methods

### Recombinant expression of CD98hc

The cDNA coding sequences for the extracellular domains of human, cynomolgus, and mouse CD98hc, with mouse kappa light chain signal sequences were each cloned into a pRK5 mammalian expression vector, in a frame with C-terminal TEV-cleavable 6xHis and Avi tags (Avidity, LLC). The resulting plasmids were transfected in Expi293F™ cells (Gibco/Thermo Fisher A14527) following standard protocols. Cultures were incubated for 120 h at 37 °C, 5% CO$_2$, and 80% humidity after which supernatants were clarified by centrifugation followed by 0.2 µm filtering. The ECDs were purified using immobilized Ni-NTA chromatography, followed by size-exclusion chromatography using PBS as the running buffer. Avi-tagged CD98hc ECDs were then biotinylated using a standard BirA-500 reaction kit, according to the manufacturer's instructions (Avidity, LLC). After the reaction, biotinylated proteins were separated from excess biotin and BirA by an additional round of size-exclusion chromatography, again using PBS as the running buffer. The proteins were then concentrated to >5 mg/ml, flash frozen in liquid nitrogen, and stored at −80 °C.

### Phage display library generation and panning

Library generation and phage display were performed according to established protocols[60]. Phage libraries were designed as described in the supplementary methods and cloned into a display plasmid containing C-terminal 6xHis and c-Myc tags and a truncated P3 protein from M13 phage. To select for huCD98hc-binding clones, biotinylated huCD98hc ECD was immobilized on streptavidin magnetic beads (Invitrogen, 11206D). After immobilization, beads were washed with 1×PBS containing 1% BSA and 0.05% Tween-20 (used for all subsequent wash steps) to remove unbound huCD98hc and further blocked with 1% BSA for at 1 h at room temperature. HuCD98hc immobilized beads were then incubated with phage libraries in 1xPBS with 1% BSA for 1 h at room temperature, followed by three, 1-min washes. Bound phage were eluted with 0.1 M glycine pH 2.7 and neutralized with 1 M Tris-HCl, pH8.0. Four rounds of selection were performed and in each subsequent round, the concentration of soluble biotinylated huCD98hc was reduced and the time of washing was prolonged to increase selective pressure.

### Yeast display library generation and sorting

DNA coding for the wild-type human Fc was incorporated into a yeast display vector as a C-terminal fusion to the Aga2p cell wall protein. An N-terminal c-Myc tag was added between the Fc and Aga2p to facilitate detection of surface-expressed protein, leaving a free C-terminus of the Fc. Yeast libraries were designed as described in the supplementary methods. To generate each library, two PCR reactions were performed using oligos with degenerate codons as outlined in the supplementary methods. These reactions were purified, mixed, and used as templates for a second PCR containing amplification primers with 50 base-pair homology to the display vector. Libraries were generated according to established protocols[61].

FACS selections were performed similarly as described[12]. Briefly, libraries were labeled with chicken anti-cMyc (1:1000, Invitrogen A-21281) and anti-chicken Alexa Fluor 488 (1:1000, Invitrogen A-11039) to assess library expression on the surface of yeast and with biotinylated huCD98hc (concentration used depended on sort round). Singlet yeast showing high binding to huCD98hc, as detected using neutravidin DyLight 650 (1:1000, Invitrogen 84607), were isolated using a FACS Aria III cell sorter, propagated in selective media, and induced for further analysis or sorting. Decreasing concentrations of soluble biotinylated huCD98hc across sort rounds was used to increase selective pressure and select for variants with improved affinity to huCD98hc. Each sort round interrogated 10x the number of clones collected in the prior round to ensure coverage of library diversity.

## Recombinant expression of TVs and antibodies

The light and heavy chain sequences of antibodies and TV variants were cloned into a pRK5 expression vector, using anti-DNP[47] or anti-BACE1[53] Fabs, as specified, and expressed in Expi293F™ cells (Gibco/Thermo Fisher A14527), Horizon CHO, or CHOK1 GS knock-out cells using standard methods. In some cases, as indicated, cloned sequences also encoded mutations for the desired effector attenuating function L234A, L235A[62] and P329G[63], (EU numbering) TV variant, or knob/hole[64] combinations. Recombinant ATV variants were subsequently affinity purified from clarified culture supernatants using protein A chromatography (Mab Select SuRe, Cytiva), followed by size-exclusion chromatography (Superdex 200, Cytiva), and stored in PBS at 4 °C or 10 mM sodium acetate with 6% sucrose, pH5.5 as previously described[12]. The identity of purified, intact ATV molecules was confirmed by LC/MS, and a purity of >95% was confirmed by SDS-PAGE and analytical HPLC-SEC.

## Affinity measurements by surface plasmon resonance

Affinities of ATVs for CD98hc were determined by surface plasmon resonance using a Biacore™ 8 K instrument. ATVs were immobilized on a Cytiva Series S CM5 sensor chip (Cytiva, 29149603) using a Cytiva Human Fab capture kit (Cytiva, 28958325) at 10 μg/mL using a flow rate of 10 μL/min for 60 s. 3-fold serial dilutions of recombinant CD98hc at concentrations of 24.5, 74.0, 222, 667, 2000, and 6000 nM were injected at a flow rate of 30 μL/min for 300 s followed by a 600-s dissociation in a 1X HBS-EP+ running buffer (Cytiva, BR100826). Data analysis was conducted using Biacore Insight Evaluation software (version 2.0.15.12933). For affinities tighter than 600 nM a kinetic analysis was performed with a 1:1 Langmuir kinetic binding model for evaluation of $k_{on}$, $k_{off}$ and $K_D$. Affinities weaker than 600 nM were measured 3 independent times and averaged using a steady state model analysis to determine $K_D$ values.

## Cell binding/uptake

Cell binding was assessed similarly as described[12]. Briefly, HeLa cells (ATCC, CCL-2), which endogenously express CD98hc, CHO-K1 cells stably expressing cynomolgus CD98hc (ChemPartner, custom order), or CHO parent cells (ChemPartner, item discontinued) were plated in triplicates at 15,000 cells/well in a 96-well Poly-D lysine-coated plate (Fisher Scientific, Perkin Elmer LLC 6055302) and incubated overnight at 37 °C. Cells were treated with ATV^CD98hc:DNP molecules, a positive control anti-CD98hc antibody[65], or anti-DNP negative control antibody for 1 h at 37 °C, fixed with 4% paraformaldehyde, and blocked with 1xPBS containing 5% BSA and 0.3% Triton. Cells were then incubated with anti-human IgG AlexaFluor®488 (1:1000, Jackson ImmunoResearch, 109-545-003), DAPI, and Deep Red Cell Mask (ThermoFisher, C10046). A minimum of 20 field of views were acquired for each replicate at ×40 using the Opera Phenix High Content imaging system (PerkinElmer). Images were analyzed using the Harmony Software (PerkinElmer version 4.9).

## Crystal structure of TV6.6-CD98hc

The TV6.6 and CD98hc complex was crystallized by sitting drop vapor diffusion, where 200 nL protein solution at 9.52 mg/mL was mixed with 200 nL mother liquor (10% Glycerol, 0.1 M HEPES pH 7.5, 5% PEG 3000, 30% PEG 400) over a reservoir of mother liquor. X-ray diffraction data was collected at the Canadian Light Source CMCF-ID beamline with an X-ray wavelength of 1.18071 Å at 100 K. Data were integrated and scaled with XDS v20220110[66] and phases were obtained by molecular replacement with Phaser v2.8.3[67]. The molecular model was built with COOT v0.9.8.7[68] and refined in Refmac5 v5.8.0403[69] to Rwork/Rfree values of 0.204/0.245. Final Ramachandran statistics were 96.42% favored, 3.42% allowed, 0.16% outliers. Solvent accessible and buried surface area was calculated in with PyMol v2.5.2[70].

## Animal care

All procedures in animals were performed in adherence to ethical regulations and protocols approved by Denali Therapeutic Institutional Animal Care and Use Committee. Mice were housed under a 12-h light/dark cycle and had access to water and a standard rodent diet (LabDiet 5LG4, Irradiated) ad libitum. Temperature and humidity in all animal rooms were monitored daily by Thermo Scientific™ InSight. The normal temperature range was 18.3–23.3 °C and the normal humidity range was 30–70%.

## In vivo mouse studies

**Dosing, blood, and tissue collections.** All mouse studies utilized males and females in the C57Bl6J background (*Mus musculus*, Ozgene, The Jackson Laboratory) between 1.2 and 5.0 months of age. Animals were approximately equally distributed between treatment groups based on sex and age. Mice were IV dosed via the tail vein with 50 mg/kg of test articles. In repeat dosing studies, animals were dosed once weekly for 4 weeks (5 doses). For in-life submandibular or submental bleeds, <50 μL of blood was collected per time point sampled, and mice received no more than two in-life blood draws within a week period and no more than seven total in-life bleeds over a four-week period. At terminal sacrifice, blood (~500 μL) was collected via cardiac puncture. Blood was collected in EDTA tubes to prevent clotting and spun at 17,200g for 7 min to isolate plasma, which was diluted 1:2000 and 1:20000 for [huIgG] analysis. Animals were deeply anesthetized via IP injection of 2.5% Avertin for terminal collections. After blood collection, mice were transcardially perfused with PBS. Fresh-frozen brain tissue was homogenized using a Qiagen TissueLyser with 5 mm steal beads for 6 min at 30 Hz in 10x tissue weight of lysis buffer containing 1% NP-40 in PBS with protease inhibitors. Brain lysates were diluted 1:2 and 1:20 for analysis of huIgG concentration.

**Quantification of huIgG.** HuIgG concentrations were quantified using a generic anti-human IgG sandwich-format ELISA. Briefly, plates were coated overnight at 4 °C with donkey anti-human IgG (JIR #709-006-098) at 1 μg/mL in sodium bicarbonate solution (Sigma #C3041-50CAP) with gentle agitation. Plates were then washed 3x with wash buffer (PBS + 0.05% Tween 20). Assay standards and samples were diluted in PBS + 0.05% Tween 20 and 1% BSA. Standard curve preparation ranged from 0.41 to 1500 ng/mL or 0.003 to 10 nM (BLQ < 0.03 nM). Standards and diluted samples were incubated with agitation for 2 h at room temperature. After incubation, plates were washed 3× with wash buffer. The detection antibody, goat anti-human IgG (JIR #109-036-098), was diluted in blocking buffer (PBS + 0.05% Tween-20 + 5% BSA) to a final concentration of 0.02 μg/mL and plates were incubated with agitation for 1 h at room temperature. After a final 3× wash, plates were developed by adding TMB substrate and incubated for 5–10 min. Reaction was quenched by adding 4 N $H_2SO_4$ and read using 450 nm absorbance.

**PK calculations.** Non-compartmental analysis was estimated using Dotmatics software 5.5 (Boston, Mass) using a non-compartmental approach consistent with the intravenous route of administration. Parameters were estimated using nominal sampling times relative to the start of each administration. Samples that were below the quantitation limit (BQL) were omitted. The model linear up log down was used to calculate exposures. Descriptive statistics (mean and standard deviation) were generated using Dotmatics.

**Statistical analysis.** One-way ANOVAs and t-tests were performed using Prism 9 software (GraphPad 9.5.1). The data distribution was sufficiently normal for all experiments outlined in Supplementary Table 5, so parametric methods were used for analysis. Brain and plasma results that were missing due to falling below the lower limit of quantification were imputed as the lower limit of quantification for analysis purposes. For repeated measures experiments, analysis was performed using two-way ANOVA of experimental condition and time point. The reported F statistic and associated p-value were derived on the interaction term. For single time point experiments, analysis was performed using one-way ANOVA. Results in Supplementary Table 5 display the results for all one- and two-way ANOVA models. Additional analyses estimated pairwise differences between groups, averaged across all time points, by obtaining the Tukey Honestly Significant Different (Tukey HSD) on each ANOVA model from Supplementary Table 5 to further determine between which two groups the differences observed (Supplementary Table 6). Analyses were performed using R version 4.2.0 with base packages and emmeans version 1.8.0.

### In vivo cynomolgus monkey study
Groups of 22–48 months old female cynomolgus monkeys (*Macaca fascicularis*, n = 3/group; Charles River Laboratories) were housed at 64–84 °F, 30–70% humidity. Housing set-up is as specified in the USDA Animal Welfare Act (Code of Federal Regulations, Title 9) and as described in the Guide for the Care and Use of Laboratory Animals. Cynomolgus monkeys were IV administered a single 30 mg/kg dose of appropriate test article and euthanized 4 days post-dose and brain tissue was collected following blood collection and perfusions with PBS. One animal dosed with the biATV$^{CD98hc}$ molecule was excluded from analysis due to signs of BBB disruption. The total test article concentrations in cynomolgus monkey serum and brain lysate samples were quantified using a generic anti-human IgG sandwich-format electrochemiluminescence immunoassay (ECLIA) on a Meso Scale Discovery (MSD) platform. Briefly, 1% casein-based PBS blocking buffer (Thermo Scientific, 37528) was added to an MSD GOLD 96-well small-spot streptavidin-coated microtiter plate (Meso Scale Discovery, L45SA) and incubated for approximately 1 h. Following the plate blocking and wash steps, biotinylated goat anti-human IgG (SouthernBiotech 2049-08) at a working concentration of 0.5 μg/mL was added to coat the assay plate and allowed to incubate for 1–2 h. Subsequently, test samples were diluted (MRD of 1:100 in 0.5% casein-based PBS assay buffer) and added to the assay plate. Following the 1–2 h incubation in the capture step, a pre-adsorbed secondary ruthenylated (SULFO-TAG) goat anti-human IgG antibody (Meso Scale Discovery, R32AJ) at a working solution of 0.5 μg/mL was added to the assay plate and incubated for approximately 1 h. An assay read buffer (1X MSD Read Buffer T, R92TC) was then added to generate the electrochemiluminescence (ECL) assay signal, expressed in ECL units (ECLU). All assay reaction steps were performed at ambient temperature and with shaking on a plate shaker (where appropriate). In serum, the assay had an MRD of 100 and a dynamic calibration standard range of 19.5–2500 ng/mL in neat matrix with 8 standard points (serially diluted at 1:2 including a blank matrix sample). The brain lysate required an MRD of 50 with a dynamic range of 4.9–2500 ng/mL with a 10 standard point curve (serially diluted at 1:2 plus a blank brain lysate sample. Serum and brain lysate sample concentrations were back-calculated off the assay-specific calibration standard curve, which was fitted with a weighed four-parameter non-linear logistic regression. The sample back-calculated concentrations in ng/mL were subsequently converted to nanomolar (nM) or micromolar (μM) as the final sample results.

### Capillary depletion
After perfusion with PBS (5 mL/min for 3–6 min), brains were dissected, and the meninges and choroid plexus removed. Fresh brain was homogenized with a Dounce homogenizer in HBSS. Homogenized samples were centrifuged (1000g for 10 min), and an aliquot of the supernatant (non-cell associated fraction) was taken following an additional centrifugation at 14,000g for 10 min. Cell pellets were resuspended in 17% dextran, and an aliquot of the total cells (cell-associated fraction) was collected, washed, and lysed in buffer containing 1% NP-40 in PBS with protease inhibitors. The remaining cells, resuspended in 17% dextran were centrifuged at 4122g for 15 min. The resulting cell pellet contained vasculature and the supernatant contained parenchymal cells. The myelin layer was removed and discarded, and the remaining supernatant was transferred to a new tube and diluted with 10 mL of HBSS and spun at 4122g for 15 min yielding a cell pellet containing parenchymal cells. Both vascular and parenchymal cell pellets were lysed in buffer containing 1% NP-40 in PBS with protease inhibitors. The total protein concentrations of samples were measured using BCA. huIgG concentration was measured as described above and was normalized to the total protein concentration in the sample.

Validation of capillary depletion fractions was assessed by Western blotting. Samples were diluted to approximately 1 mg/mL of total protein, based on BCA measurements, with NuPAGE LDS sample buffer (4×) with NuPAGE sample reducing agent (10×) and boiled for 10 min. Samples were run on NuPAGE 4–12% Bis-Tris gels. To confirm pure isolation of the parenchymal fractions from mouse samples, blots were probed with antibodies against vascular markers: CLDN5 and CD31 normalized to GAPDH (Supplementary Fig. 6a, b). For cynomolgus monkey samples, blots were probed with antibodies against the vascular markers: CLDN5 and Glut1 normalized to GAPDH (Supplementary Fig. 14a, b). To assess the depletion of cell-associated proteins from the non-cell-associated fraction, blots in mouse and cynomolgus monkey were probed for TfR and CD98hc. TfR and CD98hc levels were normalized to Ponceau S staining (Supplementary Fig. 6c, d, 14c, d). See Supplementary Table 7 for details on western detection antibodies. Blots were imaged on a Li-Cor Odyssey CLx and analyzed with Image Studio (Li-cor, Version 5.2.5).

### Mouse and cynomolgus brain immunohistochemistry
For mice, after perfusion with PBS, hemi-brains were drop fixed in 4% PFA overnight. Sagittal brain sections (40 μm) were cut using a microtome (MultiBrain® Technology by NeuroScience Associates), blocked in 5% BSA + 0.3% Triton X-100, followed by fluorescent staining with primary and secondary antibodies (Supplementary Table 8) diluted in 1% BSA + 0.3% Triton X-100 and incubated overnight. Sections were mounted in Prolong glass (Thermofisher P36984).

For cynomolgus monkeys, after perfusion with PBS, the right hemisphere of the brain was collected and sectioned into 4 mm thick coronal slabs. The 4 mm thick slabs were placed into individual tissue cassettes and immersion fixed in 4% paraformaldehyde (PFA) and stored refrigerated (4 to 9 °C) for 48 h. Immediately following fixation, tissues were transferred to PBS + 0.1% sodium azide. Thick slabs were further cut coronally into 40 μm sections on a microtome (MultiBrain® Technology by NeuroScience Associates). Brain sections were photobleached in a 4.5% (w/v) $H_2O_2$ and 20 mM NaOH in PBS solution for 2 h followed by incubation in PBS under broad-spectrum LED lights for 48h (ex/Aibecy A4 Ultra Bright 25,000 Lux LED Light Box-Tracing Pad). The photobleaching protocol was adapted from ref. 71. Brain sections

were then blocked in 5% BSA + 0.3% Triton X-100, followed by fluorescent staining, as described above, with the antibodies listed in Supplementary Table 8. Sections were mounted in Prolong glass (Thermofisher P36984). Brain images were taken using a Leica SP8 Lightning confocal microscope with a ×25 water, a ×40 oil, or a ×63 oil objective with Leica Application Suite X (3.5.7.23225) software. Final images were generated in Imaris (v9.9.0, Bitplane) including generation of a colocalization overlays using the Coloc function.

## Reporting summary

Further information on research design is available in the Nature Portfolio Reporting Summary linked to this article.

## Data availability

The atomic coordinates for the TV6.6:CD98hc complex structure have been deposited in the PDB under the accession code 8G0M. Previously published crystal structures used in this study are available in the Protein Data Bank under accession codes 2DH2, 2DH3, 6IRS, 6IRT, 6JMQ, 6JMR, 6S8V, 7B00, 7DSK, 7DSL, 1HZH, and 4W4O. The mass spectrometry proteomics data have been deposited to the ProteomeXchange Consortium via the PRIDE partner repository with the dataset identifier PXD044159. Source Data are provided with this paper.

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

## Acknowledgements

The authors would like to thank Chris Koth (Denali Therapeutics) for providing helpful comments on the manuscript, Marcus Chin (Denali Therapeutics) for assistance in high-content image data extraction, and David Gonzales (UCSD) for assistance with the mass spectrometry experiments.

## Author contributions

Conceptualization: K.S.C., R.C.W., A.M., D.C., K.J.L., H.L.T., R.T., P.A., G.M.C., J.M., M.E.P., M.D., K.G., J.W.L., R.G.T., R.J.W., Y.J.Y.Z., M.S.K. Methodology: K.S.C., R.C.W., H.L.T., A.M., M.E.P., M.E.K.C. Formal Analysis: K.S.C., R.C.W., A.M., D.C., K.J.L., H.L.T., D.J.K., Y.R.-C., D.S., R.T., M.T., Y.Z., A.G., D.H., D.J., K.N.K., D.L., K.L., N.P.D.L., M.M., E.L., P.S. Investigation: K.S.C., R.C.W., A.M., D.C., K.J.L., H.L.T., J.C., D.J.K., Y.R.-C., D.S., R.T., M.T., K.X., A.Y., Y.Z., L.A., K.B., M.B., G.M.C., T.E., D.H., D.L., A.W.-S.L., I.B., M.M., H.N.N., E.L., E.R., P.S., M.E.K.C., J.D. Writing—Original Draft: K.S.C., R.C.W., A.M., Y.J.Y.Z., M.S.K. Writing—Review & Editing: K.S.C., R.C.W., A.M., D.C., D.L., N.P.D.L., M.E.P., K.G., J.W.L., R.M., R.G.T., R.J.W., Y.J.Y.Z.,

M.S.K. Visualization: K.S.C., R.C.W., A.M., N.P.D.L., Y.J.Y.Z., M.S.K. Supervision: R.D., M.D., J.D., K.G., J.W.L., C.M., R.M., H.S., R.G.T., R.J.W., Y.J.Y.Z., M.S.K. Project Administration: K.S.C., R.C.W., J.M., Y.J.Y.Z., M.S.K.

## Competing interests

K.S.C., R.C.W., A.M., D.C., K.J.L., H.L.T., J.C., D.J.K., Y.R.C., D.B.S., R.K.T., M.T., K.X., A.Y., Y.Z., P.A., L.A., G.M.C., T.K.E., A.G., D.H., D.J., K.N.K., D.L., A.W.S.L., K.W.L., N.P.D.L., I.B., J.M., H.N.N., E.I.L., M.E.P., E.R., P.S., M.E.K.C., M.S.D., J.D., K.G., J.W.L., C.S.M., R.M., H.S., R.G.T., R.J.W., Y.J.Y.Z., M.S.K. are paid employees of Denali Therapeutics Inc. Denali has filed patent application no. PCT/US2022/053220 related to the subject matter of this paper, which includes the discovery and application of the CD98hc TVs. K.S.C., R.C.W., H.L.T., P.A., G.M.C., M.S.D., Y.J.Y.Z. and M.S.K. are inventors of this patent application. There are no other competing interests.
