## [Peer Review File · Nature Communications]

REVIEWER COMMENTS

Reviewer #1 (Remarks to the Author):

The manuscript 'Targeting CD98hc for brain delivery of biotherapeutics' by Chew et al describes the development of a brain shuttle utilizing the CD98 protein expressed at the BBB. The manuscript is easy to read, contains extensive experimental data and will be of great interest to the BBB field. The conclusions are supported by the presented data. However, some changes/additions would further increase the impact of the paper.

1. Introduction: It is true that the delivery of antibodies to the brain w/o a shuttle is very low which probably hampers the efficacy of brain targeted (immuno)therapies. Still, anti-A β antibodies w/o a brain delivery platform have shown reduction in brain A β and decreased cognitive decline (=efficacy). This should be acknowledged.
2. Abstract/Introduction/Discussion: The authors could add a bit more on the benefits of this ATV, not just stating that it is different from ATVs binding to TfR. What diseases could benefit from the slower and non-neuronal delivery characteristics?
3. On the same topic. The authors state in the Discussion that "...ATV(CD98hc) has lower peripheral clearance, prolonged brain exposure, and slower trafficking kinetics.....relative to ATV(TfR)". These characteristics are most likely good from a therapeutic perspective, but the authors should add how the total exposure and fold-increase compared to an IgG (or similar) is for ATV(CD98hc) versus ATV(TfR). Simply, which ATV is producing the highest improvement compared to a normal IgG in terms of delivery across the BBB?
4. The affinity range of the tested TVs was large, but none of the versions showed sub-nanomolar affinity. Why was no high affinity version tested? Would it not have been a good strategy to show that indeed moderate affinity is preferable?
5. It is sometimes a bit unclear what affinity version was used in the different experiments (repeated dosing, monkey) and this should be clarified especially in the methods section. For example, was the affinity comparable between the mouse and monkey given that a particular variant in general showed lower affinity towards the monkey CD98? Or was the same variant used, meaning that the affinity was lower in the monkey?
6. Why is the brain elimination of ATV(CD98) slower than that of ATV(TfR)? The authors explain that the ATV(CD98):BACE is eliminated faster than the ATV(CD98):DNP due to the former's intraneuronal distribution and lysosomal degradation. Do the authors think that this also is an explanation for the differences between ATV(TfR) and ATV(CD98)? What is the elimination route for 'unbound' ATVs in parenchyma, i.e. the fraction that is not distributed into the neurons or glia?
7. Did the authors quantify how much of the ATV(CD98) that bound to glia? And how much of the ATV(CD98):BACE that bound to glia?
8. Why did the authors decide to use the ATV:DNP and not the ATV:BACE in the cynomolgus monkey? In the end, the ATV will not be used on its own but rather to shuttle other proteins across the BBB. Considering the rather large difference in brain exposure between the ATV:DNP and ATV:BACE in the mice, it would have been more relevant to study the larger BACE-construct in the monkey. If these data are not available this should be acknowledged in the Discussion
9. What was the age and sex of the mice? And of the monkeys?
10. What was the rationale behind the (rather high) 50 mg/kg dose? For comparison, anti-A β antibody aducanumab is doses IV at 10 mg/kg every 4 weeks.

11. Minor: the authors mention that one cynomolgus monkey was excluded from the analysis due to signs of BBB disruption. Was this related to the ATV?
12. Minor: Sometimes it looks like part of the text is formatted in a different font/size, e.g. compare line 619 and 620 on page 23 (also at several other locations)

Reviewer #2 (Remarks to the Author):

In this work the authors explore the pharmacokinetics of a transport vehicle (TV) analogous to their previously reported TVs that target transferrin receptor, only this case they are targeting CD98hc (which they have also previously reported). As such, novelty is moderate, but the results are of interest to the BBB field and executed at a high level - the protein engineering component of this work is efficiently and elegantly well done.

- 1) The work demonstrates that a CD98hc TV exhibits “differentiated brain delivery with markedly slower and more prolonged kinetic properties ” than a TfR TV. Given the breadth and depth of expertise of this group in this field, one could hope for a more critical analysis of these results, particularly in comparison to the currently more-pursued TfR. Is CD98hc a better TV target than TfR? Under what circumstances? Is it worse? For which cases?
- 2) It is found that “ATVCD98hc cell-specific biodistribution in the CNS is non-neuronal, restricted to subsets of glial cells.” Isn’t this problematic if delivery to neuronal targets is desired?
- 3) “binding to either CD98hc or TfR can enable enhanced brain uptake compared to control hulgG, although each has a distinct kinetic profile. ” The neutral tone of this comment is surprising, since disappearance of ATV-TfR after 7 days is in most cases going to be far less desirable than ATV-CD98hc’s persistence for several weeks. Most indications for these targeted antibodies are likely to be chronic rather than acute.
- 4) 6-12 -fold increases in brain exposure (corresponding to 20-40nM in the brain compartment) are attained for several weeks in humanized CD98hc mouse models, certainly a significant improvement. By contrast, the isotype control only reaches 1-5 nM concentration. One wonders, though – if the EC50 (or Kd) of a therapeutic antibody were, for example, 50 pM (generally attainable with state of the art antibody engineering), then the TV would increase exposure from 20-100x to 400-800x the EC50. Would this provide a functionally meaningful improvement, or would PD effects be fully saturated in both cases?
- 5) Perhaps the distinction in persistence and cellular distribution between ATV-TfR and ATV-CD98hc indicates design tradeoffs for all ATV’s: achievement of elevated total brain concentration, at the expense of either accelerated TMDD or sequestration on untargeted cell types. Is this framing correct? The reader would benefit from the authors’ insights on these design principles.

Reviewer #3 (Remarks to the Author):

This is a well-written manuscript that clearly establishes CD98hc as a valid target to achieve high exposure of protein therapeutics in the brain via exploiting the RMT pathway. It also compares the performance of CD98hc targeted transport vehicles (TVs) with TfR targeted TV, and provides distinction between the two pathways. Inclusion of a wide area of research including protein design, characterization, PK studies and mathematical modeling work, make the manuscript very robust and worth publishing. Below are a few comments to help improve the quality of the manuscript.

-If one assumes that only free non-cell associated concentrations are pharmacologically active, it will be good to know how CD98hc affinity affects that, and if there is a 'sweet spot' for these concentrations. Maybe authors can multiply absolute concentrations reported in Figure 3C, Supp Figure 64 and Supp Figure 6g with non-cell associated fractions reported in Figure 5f, Supp Figure 6f and Supp Figure 6h, to derive such concentrations and see if there is an impact of affinity on these concentrations.

-The comparison with TfR is seminal. How did the author arrive at 620 nM KD TfR TV for comparison with 150 nM CD98hc TV. Would a different affinity or affinities change the comparison made by them? It is important to dwell on this topic.

-Use of a fit-for-purpose model to capture the PK data and conduct the pathway analysis in Figure 5 is commendable. Would it be possible to add Normal IgG and non-targeted TV data here as well to see how the flux changes in magnitude and time for those molecules with no RMT.

- It would have been prudent to collect other tissues expressing CD98hc from mouse and measure the concentration of TVs in them as well to help assess clinical translatability of safety findings. We all know that mice are more tolerant than humans. However, if the Fab domain binds to a target also expressed in skin, and if TVs accumulate in skin, which has one of the highest CD98 expression, one may get on-target off-site toxicity.

-Please discuss what is relative cellular internalization and transcytosis rates for CD98hc and TfR across BBB. Also, how the level of expression compares between them. The notion of 'TfR gives you shorter Tmax and CD98hc gives you higher AUC' should be supported using these molecular parameters and would be good to discuss in the discussion section.

- Minor: What was the KD of anti-CD98hc mAb mentioned on line 411, and how it compares with the KD of TVs? This will provide an idea about relative target engagement and safety relevance of clinical data generated using the mAb.

Reviewer #4 (Remarks to the Author):

The paper describes a new target for delivery of cargo across the BBB, through receptor-mediated transcytosis with CD98hc (SLC3A2) and compares this to the Tfr pathway. The paper is of great importance as there is a very strong need for improving Ab uptake to the brain. The CD98hc gives slower concentration-time profiles than Tfr-mediated uptake and indicates a promising pathway to steer to different cellular structures within the brain (explained on line 401-404).

The paper is well-written. It also addresses the pharmacokinetics of the transported cargo, something that is in great need instead of only using %Injected Dose. The authors actually show that using %ID would have given erroneous conclusions if used (lines 396-98).

Some minor comments:

- Line 386-88 "Importantly, consistency in enhanced brain uptake and biodistribution of ATVCD98hc was observed in cynomolgus monkey provides support": Something language-wise lacking?
- Line 529: Where were the i.v. doses administered? How many blood samples were taken from each animal? Volume?
- Line 612: Perfusion with PBS - what volume? Time?
- Line 639 and 658: It seems strange to add tables within the text that are not numbered.
- Supplemental Table 4: Some of the parameters have a very large CV, but at this stage it may be ok.

RESPONSE TO REVIEWERS

Reviewer #1 (Remarks to the Author):

The manuscript ‘Targeting CD98hc for brain delivery of biotherapeutics’ by Chew et al describes the development of a brain shuttle utilizing the CD98 protein expressed at the BBB. The manuscript is easy to read, contains extensive experimental data and will be of great interest to the BBB field. The conclusions are supported by the presented data. However, some changes/additions would further increase the impact of the paper.

We thank the reviewer for commenting that the manuscript will be of great interest to the BBB field and are happy the reviewer feels the data presented supports the conclusions made. We are appreciative of the reviewer’s insightful comments, which have helped strengthen the paper.

1. Introduction: It is true that the delivery of antibodies to the brain w/o a shuttle is very low which probably hampers the efficacy of brain targeted (immuno)therapies. Still, anti-A β antibodies w/o a brain delivery platform have shown reduction in brain A β and decreased cognitive decline (=efficacy). This should be acknowledged.

We thank the reviewer for making this point and agree it is important to explicitly acknowledge in the introduction given the recent positive clinical data and approvals for anti-Abeta. However, we believe there is still room for improvement to achieve not just higher exposure but also broader parenchymal biodistribution that would enable improved target engagement to drive better efficacy. We have edited and added to the first paragraph of the **Introduction** to further highlight these points (new text in red):

Only 0.01-0.1% of circulating antibodies cross the BBB, which despite the recent clinical success of anti-amyloid drugs [Haeberlein et al 2022, Dyck et al 2022], is typically insufficient to drive the target engagement required for efficacy. Limited BBB transport of antibodies is likely due to a combination of low vesicular trafficking and preferential lysosomal degradation of internalized antibody in brain endothelial cells [Villasenor et al. Sci Rep, 2016], suggesting that the predominant route of entry may be via the blood-CSF barrier. Subsequent brain uptake relies mainly on diffusion and distribution therefore may often be effectively limited to brain surfaces and the perivascular spaces. Increasing brain exposure while enhancing biodistribution of biotherapeutics throughout the brain parenchyma could substantially increase target engagement and lead to further improvements in efficacy.

We have also added some further language later in the **Introduction** to provide additional justification for using an RMT approach to achieve a more universal distribution within the CNS:

Molecules entering the CNS via BBB-expressed receptors take advantage of the extensive vascularization of the brain to deliver drugs throughout the parenchyma.

2. Abstract/Introduction/Discussion: The authors could add a bit more on the benefits of this

ATV, not just stating that it is different from ATVs binding to TfR. What diseases could benefit from the slower and non-neuronal delivery characteristics?

We thank the reviewer for bringing up this very relevant discussion point. The slower uptake of ATV^{CD98hc} may lend itself to better enable certain classes of therapeutics relative to ATV^{TfR} ; however, it should be noted that the non-neuronal characteristic of the non-targeted platform (ATV^{CD98hc} :DNP) can be augmented by the introduction of additional binding properties to the molecule. We show this in the ATV^{CD98hc} :BACE1 **Results** section, where in Figure 5, we describe how anti-BACE1 Fabs can localize ATV^{CD98hc} :BACE1 to neurons and elicit a pharmacodynamic response. This feature of the CD98hc platform enables targeting of different cell types within the brain parenchyma.

We agree with the reviewer that it is important to expand on how the two platforms can be best used. Rather than focusing on specific diseases, we elected to expand the **Discussion** to include language around what classes of targets each of the two platforms may best enable (edits/additions in red):

The availability of **the** complementary TV^{TfR} and TV^{CD98hc} platforms provides an opportunity to tailor platform selection based on cell type-specific therapeutic targeting and desired mechanisms of action. TV^{TfR} may be preferred for therapeutic targets that require cellular **internalization, targeting within the endolysosomal pathway (e.g., enzyme replacement therapy for lysosomal storage diseases), or therapeutics** that rely on achieving rapid C_{max} . TV^{CD98hc} may be well suited for **targets that are extracellular, on the cell surface requiring extended target engagement (e.g., antagonists), glial-specific, requiring FcγR engagement, or would benefit from sustained** brain exposure.

3. On the same topic. The authors state in the Discussion that “...ATV(CD98hc) has lower peripheral clearance, prolonged brain exposure, and slower trafficking kinetics.....relative to ATV(TfR)”. These characteristics are most likely good from a therapeutic perspective, but the authors should add how the total exposure and fold-increase compared to an IgG (or similar) is for ATV(CD98hc) versus ATV(TfR). Simply, which ATV is producing the highest improvement compared to a normal IgG in terms of delivery across the BBB?

We thank the reviewer for commenting on how certain properties of the ATV^{CD98hc} platform could be advantageous for drug development. As pointed out, facilitating higher CNS exposures relative to an IgG (or similar) is a defining feature and we appreciate the desire to further quantify this. To ensure this comparison is easily made for ATV^{CD98hc} , fold increase over IgG is reported in the text when discussing brain uptake data, and we have now added **Supplemental Figure 13i**, which is a simulation of brain uptake, clearance, and net flux of an IgG in mouse to compliment the modeling done on ATV^{CD98hc} (Figure 5f, g). We have added language to the “*Analysis of brain PK of ATV^{CD98hc} molecules using mathematical modeling*” section of the **Results** to highlight the inclusion of this analysis for the standard antibody.

Supplemental Figure 13i: Simulated brain uptake, clearance, and total net flux in mouse for a 50 mg/kg IV dose of anti-DNP IgG.

It is not straightforward to address the question of which ATV platform, TfR or CD98hc, delivers more across the BBB. As we expand on in the discussion, a key unexpected finding of the current manuscript is that CNS exposures of CD98hc-enabled biotherapeutics are impacted significantly by CD98hc-mediated *retention*. This is important as it implies accumulation contributes in a sizeable manner to measured brain concentrations of ATV^{CD98hc} . Conversely, TfR engagement within the CNS accelerates clearance from brain (Gadkar et al, EJPB 2016). This divergence in function within the CNS makes it misleading to compare the brain concentrations of ATV^{CD98hc} and ATV^{TfR} (Figure 5b) if the aim is to determine total drug exposure. To accurately answer this question, further characterization of the ATV^{TfR} and ATV^{CD98hc} platforms would be required to understand their characteristic uptake, clearance, and net flux capacities. Furthermore, analysis of exposure profiles when targeting Fabs, enzymes or proteins are fused to each TV would need to be explored as other aspects of the molecule can modulate overall PK profiles. This would therefore encompass a significant amount of work (affinity ranges, dose ranges, time course experiments, targeting Fabs) and is beyond the scope of the present manuscript, which is focused on the development of TV^{CD98hc} . We note the importance of the overall question (TfR vs CD98hc) and are currently undertaking some work to begin to answer it. However, we envision this to be an involved set of experiments, which we believe is best left for a subsequent publication.

We do however now note this aspect in the **Discussion** and highlight the need to address it with future work:

Further work defining the uptake capacity of different RMT-based platforms is therefore necessary to understand how brain concentrations relate to overall brain exposure.

4. The affinity range of the tested TVs was large, but none of the versions showed sub-nanomolar affinity. Why was no high affinity version tested? Would it not have been a good strategy to show that indeed moderate affinity is preferable?

We thank the reviewer for raising this important point about testing stronger affinity variants. While collecting these data would be helpful in further understanding the impact of CD98hc affinity on brain uptake and distribution, technical limitations have prevented us from doing these experiments. After three rounds of affinity maturation encompassing fourteen libraries, the strongest TV^{CD98hc} has an affinity to human CD98hc of ~20nM. Despite these efforts, we note that the totality of the TV6 family is encompassed by a severely restricted sequence space. We have now added the following language to the *Engineering CD98hc-binding TVs* section of the **Results** to explicitly convey the limit reached on TV affinity:

This limited diversity in sequence space, despite significant engineering, suggests an optimal solution has been approached for the TV6 family, and that further improvements in affinity are likely challenging to achieve.

We have also added the following language to the **Discussion** to emphasize the importance of understanding the brain uptake and distribution of CD98hc binding molecules at stronger affinities, despite the inability to interrogate this question directly with the current TV^{CD98hc} platform:

Additionally, profiling brain uptake and distribution when affinities to CD98hc are significantly stronger than what has been achieved with the TV^{CD98hc} platform will be a key area for further work. Diminishing brain uptake has been observed for TfR-binding molecules with strong affinities [Yu *et al*, 2011]. Establishing whether a similar relationship exists when targeting CD98hc will be important.

5. It is sometimes a bit unclear what affinity version was used in the different experiments (repeated dosing, monkey) and this should be clarified especially in the methods section. For example, was the affinity comparable between the mouse and monkey given that a particular variant in general showed lower affinity towards the monkey CD98? Or was the same variant used, meaning that the affinity was lower in the monkey?

We appreciate the reviewer's attention to detail and agree that the affinities should be consistently provided and clarified for each of these studies. We have now added this information in all **Figure Legends** (both main and supplemental figures) where *in vivo* studies are described.

Compared to the mouse studies, which used clone 6.8, 170nM to human CD98hc, a different ATV^{CD98hc}:DNP variant was used in the monkey study (clone 6.29, 205nM to monkey CD98hc) in order to achieve a more comparable binding affinity to CD98hc across species in all studies.

6. Why is the brain elimination of ATV(CD98) slower than that of ATV(TfR)? The authors explain that the ATV(CD98):BACE is eliminated faster than the ATV(CD98):DNP due to the former's intraneuronal distribution and lysosomal degradation. Do the authors think that this also is an explanation for the differences between ATV(TfR) and ATV(CD98)? What is the elimination route for 'unbound' ATVs in parenchyma, i.e. the fraction that is not distributed into the neurons or glia?

We thank the reviewer for this very insightful question regarding what could be driving the difference in brain uptake kinetics and exposure between TV^{CD98hc} and TV^{TfR} . To better understand the cellular mechanism for this difference, we compared internalization and cellular retention/degradation of TV^{CD98hc} and TV^{TfR} *in vitro* using HEK293 cells that express both receptors endogenously. The results were consistent with the delayed brain T_{max} as well as longer brain exposure of TV^{CD98hc} , where the latter revealed slower plasma membrane internalization, compared to TV^{TfR} . Although the exact rate of internalization may differ between cell types, we believe this *in vitro* data supports our *in vivo* observations. We have now included these additional *in vitro* cell binding data as a new **Supplemental Figure 12** to further address this reviewer’s question:

Supplemental Figure 12: Trafficking of ATV^{CD98hc} and ATV^{TfR} in HEK293 cells. a. Quantification of immunocytochemical detection of huIgG in non-permeabilized HEK293 cells

immediately following a 2.5 h incubation with 250 nM of monoATV^{CD98hc}:DNP (170 nM), monoATV^{TfR}:DNP (110 nM), or anti-DNP and washout at 0 h. **b.** Representative images of cell surface huIgG immediately after washout (0 h) and later time points (**e**) in non-permeabilized cells. **c.** At designated time points following a 2.5 h incubation with 250 nM of monoATV^{CD98hc}:DNP, monoATV^{TfR}:DNP, or anti-DNP and washout at 0 h, the percent huIgG concentrations in cell lysates of permeabilized HEK293 cells were normalized to the starting concentration at washout (0 h), as measured by ELISA. **d.** Representative images of immunocytochemical detection of total huIgG immediately after washout (0 h) and later time points (**f**) in permeabilized cells. N=1 for (a) and n= 5 technical replicates (b); graphs represent mean ± SEM.

We have also added the following text to the ATV^{CD98hc} and ATV^{TfR} have distinct kinetics section of the **Results**:

The trafficking and retention of ATV^{CD98hc} and ATV^{TfR} were compared using HEK293 cells that endogenously express both TfR and CD98hc to explore potential mechanisms driving these distinct *in vivo* behaviors at the cellular level. Immunocytochemical staining of total huIgG under permeabilizing conditions revealed substantially more intracellular ATV^{TfR}:DNP compared to ATV^{CD98hc}:DNP (**Supplemental Figure 12b,e**). Conversely, non-permeabilizing staining revealed more persistent cell surface localization over time for ATV^{CD98hc}:DNP compared to ATV^{TfR}:DNP (**Supplemental Figure 12a,d, f**). Additionally, there was greater cellular retention with ATV^{CD98hc}:DNP compared to ATV^{TfR}:DNP, as measured by huIgG in the cell lysate after treatment washout (**Supplemental Figure 12c**). While the exact rate of internalization may vary in different cell types, these *in vitro* data support the *in vivo* observation that ATV^{TfR}:DNP is more rapidly internalized and degraded compared to ATV^{CD98hc}:DNP. It also provides a cellular trafficking mechanism by which ATV^{CD98hc} and ATV^{TfR} differentiate in their exposure and kinetic profiles *in vivo*.

Elimination for unbound ATV is hypothesized to be similar to untargeted control IgG (i.e., if the molecule is not bound to target, its elimination should be nonspecific). Normal IgG clearance out of the brain is believed to predominantly result from non-specific ISF turnover and drainage, ultimately to the deep cervical lymph nodes (Abbott et al., *Acta Neuropathologica*, 2018). Unbound ATV^{CD98hc} and ATV^{TfR} would thus be expected to follow this same route.

7. Did the authors quantify how much of the ATV(CD98) that bound to glia? And how much of the ATV(CD98):BACE that bound to glia?

While we agree with the reviewer that this is an interesting question, we did not quantify how much of the ATV^{CD98hc} was bound to glia as we do not feel that utilizing an analysis of our existing images from IHC would provide a way to accurately do this with respect to whole brain or even within certain brain regions. Doing so accurately would require substantially more work (e.g.

stereological sampling in a new study). This was deemed beyond scope for the present manuscript. We note that we are currently validating and optimizing new methods utilizing 3D lightsheet fluorescence imaging of whole brain that could ultimately provide such information, but this will be a much more involved set of experiments requiring substantial analysis and will best be left to a follow-up publication.

8. Why did the authors decide to use the ATV:DNP and not the ATV:BACE in the cynomolgus monkey? In the end, the ATV will not be used on its own but rather to shuttle other proteins across the BBB. Considering the rather large difference in brain exposure between the ATV:DNP and ATV:BACE in the mice, it would have been more relevant to study the larger BACE-construct in the monkey. If these data are not available this should be acknowledged in the Discussion.

The reviewer highlights an important point, that characterization of the TV^{CD98hc} platform in cynomolgus monkey was performed using non-targeting, anti-DNP Fabs. We purposefully chose to use non-targeting Fabs for this experiment so that baseline platform performance could be evaluated. We show that incorporation of additional binding properties, such as the use of anti-BACE1 Fabs, significantly alters the brain uptake, retention, and biodistribution relative to the baseline behavior of the platform in the humanized mouse model (Figure 5). Furthermore, it is expected that each therapeutic fused to the TV^{CD98hc} for brain delivery (Fabs, enzymes, proteins) will have its own unique binding and distribution profile. To avoid these confounding effects and to ensure results specific to a singular molecule were avoided (i.e., ATV:BACE1), we purposely chose to first establish the translatability of the platform from mouse to cynomolgus monkey using a non-targeted ATV^{CD98hc}.

To more clearly lay out the intent of the work in cynomolgus monkeys and as a result the rationale behind our choosing to use ATV^{CD98hc}:DNP, the following language has been added to the *CD98hc dependent brain uptake and biodistribution translates to cynomolgus monkey* section of the **Results** to partly satisfy this reviewer's request:

Non-targeting Fabs were used for this initial characterization of baseline platform performance as targeting Fabs, such as anti-BACE1, were observed to significantly influence brain uptake, clearance, and localization within the CNS in mouse (**Figure 5b, d**).

The following language in the **Discussion** has also now been edited for clarity and expanded upon, as requested by the reviewer:

Importantly, the consistency observed in the brain uptake and biodistribution of ATV^{CD98hc}:DNP between cynomolgus monkey and the humanized CD98hc^{mu/hu} KI mouse supports continued use of the mouse as a representative model for further TV^{CD98hc} characterization. However, it will be important to further confirm translatability as the TV^{CD98hc} platform is used to enable more targeted biotherapeutics.

9. What was the age and sex of the mice? And of the monkeys?

We agree these are important details to include and have updated the **Material and Methods** to include the following sets of information:

All mouse studies utilized male and female mice between 1.2 and 2.8 months of age. Animals were approximately equally distributed between treatment groups based on sex and age.

Groups of 2.6-3.6 year old cynomolgus monkeys (n=3) were IV administered a single 30mg/kg dose of appropriate test article.

10. What was the rationale behind the (rather high) 50 mg/kg dose? For comparison, anti-A β antibody aducanumab is dosed IV at 10 mg/kg every 4 weeks.

We note that we intentionally used the highest practical dose to evaluate the capacity of the CNS uptake for the CD98hc binding ATV molecules. We agree that this is a relatively high dose. Subsequent dose ranging studies will be used to characterize both systemic and CNS PK as a function of dose for the different ATV molecules. This preclinical data is needed for ultimate translation into humans, e.g. to support molecule selection, optimal dose, and dosing regimen based on specific program needs. However, such studies are beyond scope for the present manuscript.

11. Minor: the authors mention that one cynomolgus monkey was excluded from the analysis due to signs of BBB disruption. Was this related to the ATV?

We do not have any evidence to suggest that the signs of BBB disruption were related to the ATV^{CD98hc}. No other animals in the study showed similar disruption and this conclusion is significantly strengthened by the fact that loss of BBB integrity has not been observed in any of the myriad mouse studies conducted to-date with ATV^{CD98hc}.

12. Minor: Sometimes it looks like part of the text is formatted in in a different font/size, e.g. compare line 619 and 620 on page 23 (also at several other locations).

We thank the reviewer for their careful attention to detail and have properly formatted the text throughout.

Reviewer #2 (Remarks to the Author):

In this work the authors explore the pharmacokinetics of a transport vehicle (TV) analogous to their previously reported TVs that target transferrin receptor, only this case they are targeting CD98hc (which they have also previously reported). As such, novelty is moderate, but the results are of interest to the BBB field and executed at a high level - the protein engineering component of this work is efficiently and elegantly well done.

We thank the reviewer for suggesting that the manuscript will be of interest to the BBB field and are happy to hear that they believe the protein engineering work was well done. We are appreciative of the reviewer's insightful comments and queries, which we believe have helped strengthen the paper.

1) The work demonstrates that a CD98hc TV exhibits “differentiated brain delivery with markedly slower and more prolonged kinetic properties” than a TfR TV. Given the breadth and depth of expertise of this group in this field, one could hope for a more critical analysis of these results, particularly in comparison to the currently more-pursued TfR. Is CD98hc a better TV target than TfR? Under what circumstances? Is it worse? For which cases?

We thank the reviewer for bringing up this very relevant point with regards to comparing TfR and CD98hc TVs. We believe that because therapeutic targets vary widely in their biological properties (e.g., trafficking, biodistribution), tailored biotherapeutics that match appropriate mechanism of action and brain exposure profiles will be required. As a result, there is not a generically better TV platform. Rather, we believe we have uncovered truly differentiated and complimentary properties of these two TV platforms, offering the opportunity for tailored platform selection to match therapeutic target needs. We have expanded the **Discussion** section where this is currently addressed with further details on the types of targets best enabled by each platform (edits/additions in red):

The availability of **the** complementary TV^{TfR} and TV^{CD98hc} platforms provides an opportunity to tailor platform selection based on cell type-specific therapeutic targeting and desired mechanisms of action. TV^{TfR} may be preferred for therapeutic targets that require cellular **internalization, targeting within the endolysosomal pathway (e.g., enzyme replacement therapy for lysosomal storage diseases), or therapeutics** that rely on achieving rapid C_{max}. TV^{CD98hc} may be well suited for **targets that are extracellular, on the cell surface requiring extended target engagement (e.g., antagonists), glial-specific, requiring FcγR engagement, or would benefit from sustained** brain exposure.

2) It is found that “ATVCD98hc cell-specific biodistribution in the CNS is non-neuronal, restricted to subsets of glial cells.” Isn't this problematic if delivery to neuronal targets is desired?

We appreciate the reviewer highlighting this observation and presenting us with the opportunity to further clarify our thinking. The ATV^{CD98hc}:BACE1 *in vivo* study demonstrates that this

platform can localize to neurons in a Fab-dependent manner (Figure 5e) and drive a pharmacodynamic response (Supplemental Figure 11d). This is currently discussed in the ATV^{CD98hc}:BACE1 results section, where we state, “*These data suggest that a lack of baseline localization to neurons does not preclude the ability of ATV^{CD98hc} to bind to, and impact the function of, neuronal Fab targets.*”

To further emphasize and generalize this point, we have added the following sentence (in red) within the **Discussion** section when referring to this property of the platform:

We show that the addition of anti-BACE1 Fab binding to a TV^{CD98hc} drives localization to neurons, which express BACE1. **More generally, this shift to neuronal localization demonstrates that Fab target binding can redirect ATV^{CD98hc} to other cell types (e.g., neurons), even if the platform’s base case neuronal localization is minimal.**

3) “binding to either CD98hc or TfR can enable enhanced brain uptake compared to control huIgG, although each has a distinct kinetic profile.” The neutral tone of this comment is surprising, since disappearance of ATV-TfR after 7 days is in most cases going to be far less desirable than ATV-CD98hc’s persistence for several weeks. Most indications for these targeted antibodies are likely to be chronic rather than acute.

We thank the reviewer for bringing up this very important observation regarding comparison of the two platforms. As mentioned in our response to point 1) above, the selection of the optimal platform will depend on the needs of that program. For example, if high drug concentrations in the CNS soon after dosing are desired for a particular therapeutic effect, TfR may likely prove superior to the CD98hc platform; conversely, if longer duration of exposures (AUC) are desired, then the CD98hc platform may potentially prove to exhibit better performance. We expect that the specific target for the program will likely determine the preferred platform based on biodistribution and trafficking differences between TfR and CD98hc molecules. Subsequent selection of dose and dosing regimens for a particular molecule to achieve clinical success will additionally be supported by translational modeling of the preclinical data for the platforms established in this manuscript. We acknowledge these are important considerations and have expanded the **Discussion** section (as mentioned in our response to point 1) to provide more speculation on types of targets and target profiles that would be more suitable for one platform over the other.

4) 6-12 -fold increases in brain exposure (corresponding to 20-40nM in the brain compartment) are attained for several weeks in humanized CD98hc mouse models, certainly a significant improvement. By contrast, the isotype control only reaches 1-5 nM concentration. One wonders, though – if the EC50 (or Kd) of a therapeutic antibody were, for example, 50 pM (generally attainable with state of the art antibody engineering), then the TV would increase exposure from 20-100x to 400-800x the EC50. Would this provide a functionally meaningful improvement, or would PD effects be fully saturated in both cases?

This is a very interesting question posed by the reviewer. As is rightfully pointed out, the advancement of antibody engineering techniques has fostered the development of more potent immunotherapies which could substantially decrease the amount of brain entry needed to obtain

‘fully saturated’ PD effects. However, it should be noted that the nature of the target is a key determinant of this. Specifically, low-density targets with slow turnover may not necessitate high concentrations of antibody in brain to achieve some level of a pharmacodynamic response. Conversely, a target that is modestly expressed with marginal turnover will likely require significantly increasing amounts of antibody to maintain target engagement over time, regardless of the antibody’s affinity for the target.

More importantly, there are critical biodistribution differences that are not adequately captured by the brain lysate concentrations measured for RMT-enabled molecules like the TV and the isotype control that the reviewer references for this thought exercise. Simply put, our data demonstrates that the TV crosses the BBB to yield a fairly homogenous distribution across different brain regions while the weight of evidence suggests isotype control antibodies predominantly access the brain via the cerebrospinal fluid after first crossing the blood-CSF barriers, which are more permissive to non-targeted IgG transport than the BBB. Limited BBB transport of non-targeted IgG is due to a combination of low vesicular trafficking and preferential lysosomal degradation of internalized IgG in brain endothelial cells (Villasenor et al. Sci Rep, 2016). The end result is that isotype control antibodies (as well as current therapeutic antibodies) are not distributed homogeneously within the brain, resulting in much higher superficial brain concentrations while deeper brain regions further from the CSF compartment are expected to see very minimal exposures, if any. There will certainly be cases where potent immunotherapies can achieve some level of therapeutic efficacy by themselves. The recent clinical successes of anti-Abeta antibodies nicely provide examples of this (note that we have now added language in the **Introduction** speaking to these examples), but their expected heterogenous biodistribution likely means that there is considerable room for improvement. We have now edited and expanded upon the first paragraph of the **Introduction** to more fully describe the biodistribution limitation for antibodies not engineered to undergo RMT at the BBB (edits/additions in red):

Only 0.01-0.1% of circulating antibodies cross the BBB, which **despite the recent clinical success of anti-amyloid drugs [Haeblerlein et al 2022, Dyck et al 2022]**, is typically insufficient to drive the target engagement required for efficacy. **Limited BBB transport of antibodies is likely due to a combination of low vesicular trafficking and preferential lysosomal degradation of internalized antibody in brain endothelial cells (Villasenor et al. Sci Rep, 2016), suggesting that the predominant route of entry may be via the blood-CSF barrier. Subsequent brain uptake relies mainly on diffusion and distribution therefore may often be effectively limited to brain surfaces and the perivascular spaces. Increasing brain exposure while enhancing biodistribution of biotherapeutics throughout the brain parenchyma could substantially increase target engagement and lead to further improvements in efficacy.**

In summary, we believe the more uniform, homogenous biodistribution of the TV due to RMT at the BBB would indeed be expected to provide a functionally meaningful improvement in PD responses even when matched against a highly potent therapeutic antibody.

5) Perhaps the distinction in persistence and cellular distribution between ATV-TfR and ATV-CD98hc indicates design tradeoffs for all ATV's: achievement of elevated total brain concentration, at the expense of either accelerated TMDD or sequestration on untargeted cell types. Is this framing correct? The reader would benefit from the authors' insights on these design principles.

We thank the reviewer for questioning the necessary trade-offs associated with leveraging RMT for brain delivery. Trafficking receptors are often overexpressed at the BBB as their function is to provide the brain with critical biomolecules that cannot passively gain access to the CNS. Importantly, the brain is not the only organ with such requirements, many times resulting in broad expression of the same trafficking receptors peripherally. TfR and CD98hc are both examples of receptors highly expressed at the BBB while also having significant expression throughout the body (but low expression on peripheral endothelial cells). As the reviewer points out, targeting such receptors enables brain delivery through RMT but comes at the price of accelerated TMDD. The nature of TMDD in the periphery will be reliant upon innate properties of each receptor, namely expression level and trafficking kinetics.

We have performed cell trafficking experiments to expand on the differences in the kinetics of CD98hc and TfR internalization, which can help explain the divergent behavior of the two platforms *in vivo*. These data (the new **Supplemental Figure 12**) illustrate that TfR has a faster internalization rate compared to CD98hc, which exhibits prolonged retention on the plasma membrane, offering a cellular mechanism for the relative TMDD rates observed *in vivo*. The following text was added to the **Results** section further discussing these data:

The trafficking and retention of ATV^{CD98hc} and ATV^{TfR} were compared using HEK293 cells that endogenously express both TfR and CD98hc to explore potential mechanisms driving these distinct *in vivo* behaviors at the cellular level. Immunocytochemical staining of total huIgG under permeabilizing conditions revealed substantially more intracellular $ATV^{TfR}:DNP$ compared to $ATV^{CD98hc}:DNP$ (**Supplemental Figure 12b,e**). Conversely, non-permeabilizing staining revealed more persistent cell surface localization over time for $ATV^{CD98hc}:DNP$ compared to $ATV^{TfR}:DNP$ (**Supplemental Figure 12a,d, f**). Additionally, there was greater cellular retention with $ATV^{CD98hc}:DNP$ compared to $ATV^{TfR}:DNP$, as measured by huIgG in the cell lysate after treatment washout (**Supplemental Figure 12c**). While the exact rate of internalization may vary in different cell types, these *in vitro* data support the *in vivo* observation that $ATV^{TfR}:DNP$ is more rapidly internalized and degraded compared to $ATV^{CD98hc}:DNP$. It also provides a cellular trafficking mechanism by which ATV^{CD98hc} and ATV^{TfR} differentiate in their exposure and kinetic profiles *in vivo*.

With respect to the reviewer's question around sequestration on non-target cell types (which we are interpreting as $ATV^{CD98hc}:DNP$ localization to astrocytes/microglia but not neurons), we show that ATV^{CD98hc} preferentially associates with astrocytes and microglia, depending on valency and

FcγR binding. However, we do not believe these distribution patterns to be the only ones possible. For example, though non-targeting ATV^{CD98hc}:DNP does not show significant localization to neurons, appendage of anti-BACE1 Fabs results in neuronal uptake with a corresponding pharmacodynamic response (Figure 5d, Supplemental Figure 11d). We therefore believe a feature of the CD98hc platform is that, given the rest of the molecule is designed appropriately, all the major cell types within the brain parenchyma can be targeted.

Reviewer #3 (Remarks to the Author):

This is a well-written manuscript that clearly establishes CD98hc as a valid target to achieve high exposure of protein therapeutics in the brain via exploiting the RMT pathway. It also compares the performance of CD98hc targeted transport vehicles (TVs) with TfR targeted TV, and provides distinction between the two pathways. Inclusion of a wide area of research including protein design, characterization, PK studies and mathematical modeling work, make the manuscript very robust and worth publishing. Below are a few comments to help improve the quality of the manuscript.

We thank the reviewer for commenting on the breadth of the work included in the manuscript and are grateful that they agree the work is worthy of publication. We are further appreciative of their insightful comments, which have helped strengthen the paper.

1) If one assumes that only free non-cell associated concentrations are pharmacologically active, it will be good to know how CD98hc affinity affects that, and if there is a ‘sweet spot’ for these concentrations. Maybe authors can multiply absolute concentrations reported in Figure 3C, Supp Figure 64 and Supp Figure 6g with non-cell associated fractions reported in Figure 5f, Supp Figure 6f and Supp Figure 6h, to derive such concentrations and see if there is an impact of affinity on these concentrations.

We thank the reviewer for questioning the interpretation of cell associated vs non-cell associated fractions of ATV^{CD98hc} in brain and appreciate the opportunity to expand upon this. We show that a fraction of the non-targeting ATV^{CD98hc} :DNP in brain remains cell associated in an affinity-dependent manner; stronger affinity leads to higher association. In this situation, we agree with the reviewer that these ATVs *could* be pharmacologically inactive as they may be sequestered on the cell surface. However, it is important to note that this observation is made on the baseline platform with non-targeting Fabs. To evaluate whether the cell-associated fraction can be functionally active in a therapeutic context, we point to the results obtained with ATV^{CD98hc} :BACE1 (Figure 5d, e and Supplemental Figure 6). Here, anti-BACE1 Fabs were appended to the ATV^{CD98hc} platform and subsequently shown to localize to neurons and elicit a pharmacodynamic response. The critical point here is that the baseline behavior of the platform is non-neuronal; however, with the addition of Fabs that bind to a neuronal target, the molecule can be redirected to a different cell type within the brain parenchyma. We believe these data highlight that the observed biodistribution and cell associated nature of the ATV^{CD98hc} :DNP is easily overcome in a therapeutic context, and as such, do not view cell-associated fractions as likely to limit pharmacological activity for these molecules.

2) The comparison with TfR is seminal. How did the author arrive at 620 nM KD TfR TV for comparison with 150 nM CD98hc TV. Would a different affinity or affinities change the comparison made by them? It is important to dwell on this topic.

We thank the reviewer for highlighting the importance of the comparison between TfR and CD98hc enabled TVs and appreciate the need to understand how these clones were chosen for the head-to-head study. In our previous publication (Kariolis et al, *Sci Transl Med* 2020), we show that peripheral and brain PK for ATV^{TfR} variants at an affinity of ~600nM for TfR resulted in significant brain uptake while maintaining higher peripheral exposures compared to a stronger binding variant. Similarly, binding to CD98hc with a K_D of around 100 – 200nM enabled near-maximal brain uptake with prolonged plasma PK (Figure 3), though the affinity to CD98hc was less deterministic on these parameters (Supplemental Figure 6e). As a result, clones within these affinity ranges were chosen to provide the fairest head-to-head comparison of the two platforms.

It is well established that affinity to TfR is an important factor that dictates efficiency of brain uptake for molecules that bind to the receptor (Yu et al, *Sci Transl Med*, 2011). Therefore, choosing significantly stronger or weaker TfR binding ATV^{TfR} variants could impact the resulting comparison to the CD98hc platform. Gaining an understanding of this relationship across the permutations of affinities (i.e., strong and weak variants for ATV^{TfR} and ATV^{CD98hc}) would encompass a significant amount of work (affinity ranges, dose ranges, time course experiments) and is deemed beyond the scope of the present manuscript, which is focused on the development of TV^{CD98hc} . We note the importance of the overall question (TfR vs CD98hc) and are currently undertaking additional work to begin to address this at greater depth. However, we envision this to be an involved set of experiments, the analysis of which is better fit for a subsequent publication.

To provide additional clarity on why these two clones were selected for the head-to-head study, we have added the following language in the *ATV^{CD98hc}* and *ATV^{TfR}* have distinct kinetics section within the **Results** section:

These variants were chosen as these affinities to TfR and CD98hc have been shown to provide high brain exposures (Supplemental Figure 6e) while balancing peripheral target-mediated disposition [Kariolis et al, *STM* 2020].

3) Use of a fit-for-purpose model to capture the PK data and conduct the pathway analysis in Figure 5 is commendable. Would it be possible to add Normal IgG and non-targeted TV data here as well to see how the flux changes in magnitude and time for those molecules with no RMT.

We appreciate the recognition that building a mathematical PK model to further our interpretation of the *in vivo* behavior of ATV^{CD98hc} is of value and agree that providing a similar analysis for a standard IgG would be helpful. We have added the following graph of simulated uptake, clearance and net flux for a standard IgG in mouse as **Supplemental Figure 13i**.

Supplemental Figure 13i: Simulated brain uptake, clearance, and total net flux in mouse for a 50 mg/kg IV dose of anti-DNP huIgG.

4) It would have been prudent to collect other tissues expressing CD98hc from mouse and measure the concentration of TVs in them as well to help assess clinical translatability of safety findings. We all know that mice are more tolerant than humans. However, if the Fab domain binds to a target also expressed in skin, and if TVs accumulate in skin, which has one of the highest CD98 expression, one may get on-target off-site toxicity.

We agree with the reviewer that evaluation of peripheral tissue concentration is an important step towards understanding the clinical translatability of this platform. We have in fact collected this data and have now included extensive peripheral tissue pharmacokinetics data, as well as evaluation of CD98hc expression in kidney and testes which highly express CD98hc. This data is now included in the revised manuscript as **Supplemental Figure 7:**

Supplemental Figure 7

Supplemental Figure 7: Peripheral PK of mono and biATV^{CD98hc}:DNP. a-p. Peripheral tissue concentrations of anti-DNP, monoATV^{CD98hc.6.39}:DNP (94nM), or biATV^{CD98hc.10.8.d1}:DNP (165nM) after a single 50 mg/kg IV dose. Graphs represent mean \pm SD, n= 1-5/group. Missing data for select time points are due to values below the lower limit of detection. q-t. Western blot quantification of CD98hc protein expression normalized to beta actin in kidney (q, s) and testes (r, t) following a single dose 50 mg/kg IV dose of the indicated molecules at 1, 7 and 21 days post-dose. Graphs represent mean \pm SEM, n= 2-4/group, unpaired two-tailed t-test.

Although huIgG PK in skin was not collected in these studies, we did perform histopathology on skin samples in repeat dosing studies (Figure 6) and there were no significant findings attributable to the ATV^{CD98hc} molecules. We agree with the reviewer that PK in skin would be of interest in future experiments, especially if paired with a Fab target that is also expressed there. More

generally, we agree with the reviewer that Fab target expression and binding will have a profound influence on peripheral tissue uptake and safety, and each therapeutic molecule paired with TV^{CD98hc} will require independent evaluation to determine clinical translatability and safety.

5) Please discuss what is relative cellular internalization and transcytosis rates for CD98hc and TfR across BBB. Also, how the level of expression compares between them. The notion of ‘TfR gives you shorter Tmax and CD98hc gives you higher AUC’ should be supported using these molecular parameters and would be good to discuss in the discussion section.

The reviewer highlights an important point, which is that the relative expression levels of CD98hc and TfR at the BBB can have a profound influence of the brain uptake observed *in vivo*. pProtein expression levels of CD98hc and TfR on isolated human microvessels can be found on Supplemental Figure 15 of the original submitted manuscript. Importantly, levels of these two proteins were observed to be comparable in both the cortex and cerebellum:

Supplemental Figure 15

Furthermore, proteomics analysis of isolated brain endothelial cells from the mouse has been previously reported (Zuchero et al, *Neuron*, 2016; Figure 3c; see below), similarly showing CD98hc and TfR protein expressions are comparable:

Redacted Figure

Zuchero et al., *Neuron*, 2016

To address the differences in internalization and trafficking between CD98hc and TfR, we have now included additional cell trafficking time course data (**Supplemental Figure 12**) illustrating how ATV^{CD98hc} and ATV^{TfR} differ in internalization and cellular retention (e.g., degradation) and the following text in the **Results**:

The trafficking and retention of ATV^{CD98hc} and ATV^{TfR} were compared using HEK293 cells that endogenously express both TfR and CD98hc to explore potential mechanisms driving these distinct *in vivo* behaviors at the cellular level. Immunocytochemical staining of total huIgG under permeabilizing conditions revealed substantially more intracellular ATV^{TfR} :DNP compared to ATV^{CD98hc} :DNP (**Supplemental Figure 12a,b,e**). Conversely, non-permeabilizing staining revealed more persistent cell surface localization over time for ATV^{CD98hc} :DNP compared to ATV^{TfR} :DNP (**Supplemental Figure 12d, f**). Additionally, there was greater cellular retention with ATV^{CD98hc} :DNP compared to ATV^{TfR} :DNP, as measured by huIgG in the cell lysate after treatment washout (**Supplemental Figure 12c**). While the exact rate of internalization may vary in different cell types, these *in vitro* data support the *in vivo* observation that ATV^{TfR} :DNP is more rapidly internalized and degraded compared to ATV^{CD98hc} :DNP. It also provides a cellular trafficking mechanism by which ATV^{CD98hc} and ATV^{TfR} differentiate in their exposure and kinetic profiles *in vivo*.

Supplemental Figure 12: Trafficking of ATV^{CD98hc} and ATV^{TfR} in HEK293 cells. a. Quantification of immunocytochemical detection of huIgG in non-permeabilized HEK293 cells immediately following a 2.5 h incubation of monoATV^{CD98hc}:DNP, monoATV^{TfR}:DNP, or anti-DNP (250 nM) and washout at 0 h. **b.** Representative images of cell surface huIgG immediately after washout (0 h) and later time points (**e**) in non-permeabilized cells. **c.** At designated time points following a 2.5 h incubation of monoATV^{CD98hc}:DNP, monoATV^{TfR}:DNP, or anti-DNP (250 nM) and washout at 0 h, the percent huIgG concentrations in cell lysates of permeabilized HEK293 cells were normalized to the starting concentration at washout (0 h), as measured by ELISA. **d.** Representative images of immunocytochemical detection of total huIgG immediately after washout (0 h) and later time points (**f**) in permeabilized cells. N=1 for (a) and n= 5 technical replicates (b); graphs represent mean ± SEM.

These data provide a potential cellular trafficking mechanism to explain the behavior of the two TV platforms *in vivo* (i.e., why ATV^{TfR} drives a rapid C_{max} while ATV^{CD98hc} has prolonged exposure). Although these *in vitro* experiments do not provide transcytosis rates for each receptor at the BBB, which we agree with the reviewer are values of importance, the experiments required to ascertain such detailed comparative insights into *in vivo* brain delivery parameters would encompass a significant amount of work and are beyond the scope of the present manuscript. However, we appreciate the importance of the overall question and are currently planning additional lines of investigation to begin to answer it. We envision this will be an involved set of experiments, the analysis of which is better left for a subsequent publication.

6) Minor: What was the KD of anti-CD98hc mAb mentioned on line 411, and how it compares with the KD of TVs? This will provide an idea about relative target engagement and safety relevance of clinical data generated using the mAb.

We thank the reviewer for suggesting that additional detail be included describing the clinical anti-CD98hc antibody. We agree that providing this information will allow for a better interpretation of how relevant the clinical data is to what has been observed with the TV^{CD98hc} platform. We have added to and edited the cited sentence in the **Discussion** which now reads (edits in red):

This lack of observed safety findings is consistent with a phase 1 trial of an **effector positive, bivalent anti-CD98hc antibody (K_D to human CD98hc of 2 – 6 nM).**

Reviewer #4 (Remarks to the Author):

The paper describes a new target for delivery of cargo across the BBB, through receptor-mediated transcytosis with CD98hc (SLC3A2) and compares this to the Tfr pathway. The paper is of great importance as there is a very strong need for improving Ab uptake to the brain. The CD98hc gives slower concentration-time profiles than Tfr-mediated uptake and indicates a promising pathway to steer to different cellular structures within the brain (explained on line 401-404).

The paper is well-written. It also addresses the pharmacokinetics of the transported cargo, something that is in great need instead of only using %Injected Dose. The authors actually show that using %ID would have given erroneous conclusions if used (lines 396-98).

We thank the reviewer for agreeing that this manuscript encompasses work that is of great importance to the field, and for commenting on the need for in-depth pharmacokinetic analysis when interpreting brain uptake data. We are appreciative of the reviewer's insightful comments, which have helped strengthen the paper.

Some minor comments:

1) Line 386-88 "Importantly, consistency in enhanced brain uptake and biodistribution of ATVCD98hc was observed in cynomolgus monkey provides support": Something language-wise lacking?

We thank the reviewer for their careful review of our manuscript and for highlighting that this sentence was incomplete. Edits have been made to improve clarity, with the updated sentence reading as followed:

Importantly, the consistency observed in the brain uptake and biodistribution of ATV^{CD98hc}:DNP between cynomolgus monkey and the humanized CD98hc^{mu/hu} KI mouse supports continued use of the mouse as a representative model for further TV^{CD98hc} characterization.

2) Line 529: Where were the i.v. doses administered? How many blood samples were taken from each animal? Volume?

Additional details have been added to the **Materials and Methods** to address these questions:

All mouse studies utilized male and female mice between 1.2 and 2.8 months of age. Animals were approximately equally distributed between treatment groups based on sex and age. Mice were IV dosed via the tail vein with 50 mg/kg of test articles. In repeat dosing studies, animals were dosed once weekly for 4 weeks (5 doses). For in-life submandibular or submental bleeds, <50 uL of blood was collected per time point sampled, and mice received no more than two in-life blood draws within a week period and no more

than seven total in-life bleeds over a four week period. At terminal sacrifice, blood (~500mL) was collected via cardiac puncture.

3) Line 612: Perfusion with PBS - what volume? Time?

Additional details have been added to the **Materials and Methods** to address these questions:

After perfusion with PBS (5 mL/min for 3-6 minutes), brains were dissected, and the meninges and choroid plexus removed.

4) Line 639 and 658: It seems strange to add tables within the text that are not numbered.

We thank the reviewer for this comment and have added these tables as Supplemental Table 7 and Supplemental Table 8.

5) Supplemental Table 4: Some of the parameters have a very large CV, but at this stage it may be ok.

The reviewer is correct in pointing out that the CVs are on the higher end for some of the parameters in the model. As mentioned, this is an early iteration of the model and parameter estimations are across a wide range of affinity values. Due to the inherent inter-study and inter-animal variability in the data, some degree of variability in the estimates are to be expected. As additional experiments are conducted with the TV^{CD98hc} platform, the resulting data will be incorporated into the model to further refine these parameters and narrow the CVs. Nevertheless, we do feel that the model is accurate in its current form given the good agreement between it and the empirical data.

References

Abbott, N. J., Pizzo, M. E., Preston, J. E., Janigro, D. & Thorne, R. G. The role of brain barriers in fluid movement in the CNS: is there a “glymphatic” system? *Acta Neuropathol* **135**, 387–407 (2018).

Gadkar, K. *et al.* Mathematical PKPD and safety model of bispecific TfR/BACE1 antibodies for the optimization of antibody uptake in brain. *Eur J Pharm Biopharm* **101**, 53–61 (2016).

Villaseñor, R. *et al.* Trafficking of Endogenous Immunoglobulins by Endothelial Cells at the Blood-Brain Barrier. *Sci Rep* **6**, 25658 (2016).

Zuchero, Y. J. *et al.* Discovery of Novel Blood-Brain Barrier Targets to Enhance Brain Uptake of Therapeutic Antibodies. *Neuron* **89**, 70–82 (2016).

REVIEWERS' COMMENTS

Reviewer #1 (Remarks to the Author):

The authors have revised the manuscript according to both my own and my fellow reviewers' comments. It will be interesting to see further head-to-head comparisons between TfR and CD98 binding TVs (something that the majority of the reviewers commented on in one way or the other), but I have no further questions for the present paper.

Reviewer #2 (Remarks to the Author):

Revisions satisfactorily addressed this reviewer's critiques.

Reviewer #3 (Remarks to the Author):

Authors have satisfactorily addressed all the comments and I have no further improvements to suggest.

Reviewer #4 (Remarks to the Author):

The authors have responded to the posed comments in a good way. The only tiny detail is in the response to comment 2, reviewer #4. The total volume of blood collected via heart puncture cannot be 500 mL but rather 500 μ L. Correct?

RESPONSE TO REVIEWERS

Reviewer #1 (Remarks to the Author):

The authors have revised the manuscript according to both my own and my fellow reviewers' comments. It will be interesting to see further head-to-head comparisons between TfR and CD98 binding TVs (something that the majority of the reviewers commented on in one way or the other), but I have no further questions for the present paper.

We agree that additional insights into how TfR and CD98hc binding TVs compare is of great interest. We are pursuing additional experiments to this end and look forward to presenting them in a subsequent manuscript.

Reviewer #2 (Remarks to the Author):

Revisions satisfactorily addressed this reviewer's critiques.

We thank the reviewer for their suggestions throughout the review process, which have strengthened this manuscript.

Reviewer #3 (Remarks to the Author):

Authors have satisfactorily addressed all the comments and I have no further improvements to suggest.

We thank the reviewer for their suggestions throughout the review process, which have strengthened this manuscript.

Reviewer #4 (Remarks to the Author):

The authors have responded to the posed comments in a good way. The only tiny detail is in the response to comment 2, reviewer #4. The total volume of blood collected via heart puncture cannot be 500 mL but rather 500 μ L.

That is correct, the total blood collected via cardiac puncture was 500 μ L, not 500 mL. We appreciate the attention to detail and have made this correction.